# Targeted knockdown of DNA methyltransferase 3a (DNMT3a) unlocks dedifferentiation and neurogenic potential in mouse retinal Müller glia

Rebeca Victoria-Chavez [ID], Monica Lamas [ID]*

Departamento de Farmacobiología, Centro de Investigación sobre el envejecimiento, Cinvestav Sede Sur, Mexico City, Mexico

* mlamas@cinvestav.mx

## Abstract

The regenerative response of retinal cells to injury and aging depends on the epigenomic plasticity that enables the dedifferentiation and neuronal differentiation capacities of Müller glial cells (MG). In mammals, this regenerative ability is extremely limited, and disruptions in epigenetic mechanisms, particularly those involving DNA methylation and demethylation, may underlie this restricted potential. To explore this possibility, we aimed to develop DNA methylation-targeting molecular tools to enhance the dedifferentiation and neurogenic capacity of primary MG cultures derived from mouse retina. Using CRISPR/dCas9-based gene regulation technology, we selectively and transiently inhibited *Dnmt3a*, a de novo DNA methyltransferase previously implicated in maintaining transcriptional repression, for approximately 3–5 days. Our results show that *Dnmt3a* knockdown leads to sustained upregulation of pluripotency-associated genes, including *Ascl1*, *Lin28*, and *Nestin*, as measured by RT-qPCR and immunofluorescence. This epigenetic modulation also promoted increased cell proliferation and migration, both hallmarks of a regenerative response. Furthermore, *Dnmt3a* knockdown, either alone or in combination with neurogenic stimuli, induced MG to acquire neuronal-like morphologies and express the early neuronal marker βIII-tubulin. These findings suggest that *Dnmt3a* acts as a repressive regulator of MG plasticity, likely serving as an epigenetic barrier that counteracts injury-induced demethylation events. Overall, our study identifies *Dnmt3a* as a critical modulator of MG fate and highlights the potential of its targeted downregulation to facilitate reprogramming. By prolonging the transient progenitor-like state of MG, DNMT3a inhibition may serve as a complementary approach to unlock the neurogenic and regenerative potential of the mammalian retina, offering promising avenues for future therapeutic strategies.

**Data availability statement:** All relevant data are within the manuscript and its Supporting information files.

**Funding:** This work was supported by Conahcyt grant A1-S-25777 and Velux Stiftung project number 1852. A PhD fellowship from Conahcyt to R. V-C is also acknowledged. The funders had no role in study design, data collection and analysis, decision to publish, or preparation of the manuscript.

**Competing interests:** The authors have delcared that no competing interests exist.

## Introduction

Müller glia (MG) reprogramming sustains morphological and functional retinal regeneration in zebrafish [1–3]. This process is characterized by the re-entry of MG into the cell cycle, the expression of pluripotency-related genes, cell proliferation, migration and differentiation into retinal neurons [4–7]. Together, these events lead to the replacement of lost neurons following damage and contribute to the maintenance of tissue function [8–10].In mammals, the reprogramming of Müller cells is extremely limited, and neuronal regeneration does not occur [11]. However, multiple studies have shown that the reprogramming capacity of mammalian MG might be unlocked through various molecular strategies. For example, reprogramming can be induced by ectopic expression of the transcription factor Ascl1 [12–14], by disrupting intercellular Prox1 transfer in the retina in vivo [15], through activation of Notch or Wnt signaling pathways [16–19], the addition of growth factors [20], or by exposing cell cultures to high concentrations of glutamate or the glutamate analogue N-methyl-D-aspartate (NMDA) [21,22]. Many of these strategies either are accompanied or require epigenetic alterations of the genomic landscape that support reprogramming by altering gene expression signatures [14,21].Specifically, a correlation between DNA demethylation and injury-dependent pluripotency gene expression has been identified in zebrafish [23], chick [24] and mouse [25–27] DNA methylation and demethylation dynamics are also involved in cell specification or differentiation processes [28–30].

DNA methylation regulates gene expression by adding a methyl group to the C5 position of cytosine residues in DNA [31,32], primarily within CpG islands at promoter regions. This modification, resulting in 5-methylcytosine (5-mC), serves as a repressive mark for transcription. The addition is a covalent modification catalyzed by DNA methyltransferase (DNMT) enzymes [33–35].Various types of DNMTs have been identified in nearly all organisms. DNMT3a and DNMT3b are responsible for establishing DNA methylation patterns, while DNMT1 maintains these patterns during DNA replication [36,37]. DNMT3a, for example, has been identified as a crucial regulator in optic nerve regeneration [38] and its downregulation leads to a passive demethylation at important neurogenic gene promoters, enhancers and regulators sites in different cell types [39–41].

In Müller cells, distinct expression patterns of DNMTs have been observed following damage and aging in various vertebrate species [25,42]. While the role of demethylation has been explored, the specific contribution of *de novo* methyltransferases like DNMT3a in restricting mammalian MG plasticity remains unclear. It has been speculated that these differences may contribute to the activation of either a regenerative or non-regenerative response to retinal injury across species.

In this study, we found that the selective and transient transcriptional inhibition of *Dnmt3a* mediated by the CRISPRi system promotes *in vitro* Müller glia dedifferentiation. This process is associated with an increased early expression of the pluripotency-related genes *Ascl1*, *Lin28* and *Nestin*, cell migration and proliferation ratio. In addition, we report that knockdown of *Dnmt3a* induces the expression of βIII-tubulin, an early neuronal marker, both in the absence and presence of a

neurogenic stimulus such as GABA exposure [22]. Our results support the involvement of *Dnmt3a* in both the dedifferentiation and neuronal differentiation processes of mammalian Müller glial cells, potentially contributing to an epigenetic environment similar to that observed in regenerative species like zebrafish.

## Materials and methods

### Animals

Postnatal 8–12 days C57BL/6J mice were used for all experiments, they were handled and treated in strict accordance with the guidelines of the internal animal care committee (CICUAL-CINVESTAV, Project number: 0354−23).

### Statistical analysis

GraphPad Prism 8.0.1 software was used to carry out statistical analyses. Shapiro-Wilk test for normality, ANOVA or Kruskal-Wallis and Tukey pos hoc tests were performed to determine statistical differences. The data are shown as the means and error bars corresponding to the standard error of the mean (SEM). For each analysis, the value of $p < 0.05$ was considered significant.

### gRNA design

The dCas9/KRAB the vector for the cloning of gRNA was purchased from Addgene (#243314).

We identified the target sequences based on their closeness to the transcription start site (−300 to +200 bases pair) of DNMT3a gene and the presence of the protospacer adjacent motif (NGG). We choose 19–21 bp sequences for the gRNA design and checked for off target sites, non was found. The gRNA sequences are shown in Table 1.

Oligo primers were set according to the selected gRNA, then the PAM site (NGG) was virtually removed from the target sequence and 5' CACCG for sense or 5' AAAC 3'C for antisense primers were added. The sense strand and antisense strand was phosphorylated and annealed to form a double strand oligo. Then 1 µL of sense oligo (10 µM), 1 µL of antisense oligo (10 µM), 1 µL of PNK buffer with ATP, 0.5 µL of T4 polynucleotide kinase (PNK) and 6.5 µL of nuclease free water were placed in a 0.2 mL microtube and incubated at 37°C for 30 minutes, 5 minutes at 95°C, natural cooling to 25 °C, and 25°C for 20 minutes. The annealed primers were inserted into digested and dephosphorylated pLV-hU6-sgRNA-hUbC-dCas9-KRAB-T2a-Puro Vector (BsmBI and SAP reaction) using T4 DNA ligase (Jena Bioscience®), following the protocol provided by the supplier, with a 5:1 insert-to-vector ratio. Finally, 5 µL of the final ligation product was added into 50 µL of DH5α chemically competent cells put on ice for 30 min and heat chocked at 42°C for 60 seconds, and returning on ice for 2 min. 250 µL of prewarmed LB medium was added and incubated at 37°C for 90 min and constant shaking at 200 rpm. Then 150 µL of bacterial suspension was cultured in LB solid medium supplemented with 0.1 mg/mL ampicillin and incubated at 37°C for 12–16 hours. Positive clones were picked out and seeded in prewarmed LB medium supplemented with 0.1 mg/mL ampicillin and incubated at 37°C for 12–16 hours with constant shaking at 200 rpm. Plasmid extraction was performed using QIAPrep Spin Midiprep Kit (QIAGEN) according to the supplier's instructions.

**Table 1. Sequences of gRNA and target sites.**

|  | Target sequence | Sense Primer | Antisense Primer |
|---|---|---|---|
| sgDnmt3a-1 | CAAGTCCGTACAATGCCGCG | CACCGCAAGTCCGTACAATGCCGCG | AAACCGCGGCATTGTACGGACTTGC |
| sgDnmt3a-2 | AAGTCCGTACAATGCCGCGG | CACCGAAGTCCGTACAATGCCGCGG | AAACCCGCGGCATTGTACGGACTTC |
| sgDnmt3a-3 | GCCCTGCAAATAGGAAAGGG | CACCGGCCCTGCAAATAGGAAAGGG | AAACCCCTTTCCTATTTGCAGGGCC |
| sgDnmt3a-4 | AGAGGTGAGCACTACAGACC | CACCGAGAGGTGAGCACTACAGACC | AAACGGTCTGTAGTGCTCACCTCTC |

A double PCR reaction was designed to verify the identity of the plasmids. The primers used are listed in Table 2. Previously designed primers were also used as a starting point for some PCRs to confirm the ligation of the sRNA into the dCas9-KRAB vector.

## Müller glia primary cultures

Mice were euthanized and decapitated. Their heads were soaked in 70% ethanol, and the eyes were enucleated and placed in a Petri dish with cold PBS 1X, then the retinas were dissected carefully and placed in fresh cold PBS 1X. Retinas were incubated in DMEM containing 0.1% trypsin and 70 IU/ml collagenase (Sigma Chemical Co., St. Louis, MO) for 30 minutes at 37 ºC. The enzymatic digestion was stopped by adding DMEM containing 10% FBS and the retinas were dissociated through vigorous pipetting, centrifugated at 300 x g for 3 minutes and resuspended in 1 mL of media culture. The dissociated cells from 4–8 retinas were seeded into 25 cm$^2$ culture flasks with DMEM-FBS 10% and penicillin-streptomycin 1% and placed in an incubator at 37°C and 5% $CO_2$. The cells were allowed to attach for 24 hours and after, the cell cultures were washed with PBS 1X to eliminate non-adherent cells (neuronal cells), and the media was replaced. The cells were kept in culture and half of the medium was replaced every 2–3 days. The cells were maintained in these conditions until they were confluent (~80% confluence). The purity of the culture was evaluated by immunofluorescence staining for Müller cells markers and lack of expression of neuronal markers as previously described [17]. More than 95% of the cells were immunopositive to GS (not shown). Then the cells were harvested using 0.12% trypsin/EDTA (Sigma-Aldrich #T4049) and re-seeded in three 75 cm$^2$ flasks until confluent and finally re-seeded in six 175 cm$^2$ flasks until they reached ~70% confluence (cell yield of ~ 2–2.5 x10$^6$ cells/flask).

## Electroporation of müller cells

1X10$^6$ cells per transfection (on exponential growth ~70% confluency cultures) in suspension at 11.1 x 10$^6$ cells/mL in DMEM serum free were needed. We transferred 90 µL of cells suspension to a CUY21Cuvette and then 10 µL of a 1 µg/

**Table 2. List of PCR primers for ligation checking.**

|  |  | Sequence |
|---|---|---|
| *KRAB-Puro* | Forward | AGGGACAGCAGAGATCCAGTTTGGTTA |
|  | Reverse | AGATCTGGAGCCGACACGGG |
| *sgDNMT3a-1 PP1* | Forward | CACCGCAAGTCCGTACAATGCCGCG |
|  | Reverse | AGATCTGGAGCCGACACGGG |
| *sgDNMT3a-1 PP2* | Forward | AGGGACAGCAGAGATCCAGTTTGGTTA |
|  | Reverse | AAACCGCGGCATTGTACGGACTTGC |
| *sgDNMT3a-2 PP1* | Forward | CACCGAAGTCCGTACAATGCCGCGG |
|  | Reverse | AGATCTGGAGCCGACACGGG |
| *sgDNMT3a-2 PP2* | Forward | AGGGACAGCAGAGATCCAGTTTGGTTA |
|  | Reverse | AAACCCGCGGCATTGTACGGACTTC |
| *sgDNMT3a-3 PP1* | Forward | CACCGGCCCTGCAAATAGGAAAGGG |
|  | Reverse | AGATCTGGAGCCGACACGGG |
| *sgDNMT3a-3 PP2* | Forward | AGGGACAGCAGAGATCCAGTTTGGTTA |
|  | Reverse | AAACCCCTTTCCTATTTGCAGGGCC |
| *sgDNMT3a-4 PP1* | Forward | CACCGAGAGGTGAGCACTACAGACC |
|  | Reverse | AGATCTGGAGCCGACACGGG |
| *sgDNMT3a-4 PP2* | Forward | AGGGACAGCAGAGATCCAGTTTGGTTA |
|  | Reverse | AAACGGTCTGTAGTGCTCACCTCTC |

µL purified plasmid DNA suspension was added. The plasmids were transferred to the cells by electroporation using NEPA GENE 21 electroporator. Square electric pulses were applied at 275 V (pulse length, 0.5 ms; two pulses; interval, 50 ms) followed by pulses at 20 V (pulse length, 50 ms; five pulses; interval, 50 ms).

We recollected all transfected cells by pipetting and seeded in 6-well plates with DMEM/FBS 10% A/A 1% at a cell density of $8 \times 10^4$ cells per well and incubate at 37°C and 5% $CO_2$. 5 µg/mL puromycin was added 24 hours after transfection to select positive transformed cells. We confirmed the presence of the construction by immunofluorescence against Cas9 protein. After puromycin selection more than 90% of the cells were immunopositive to Cas9 (S1). Efficiency of knockdown was demonstrated by qPCR.

## Differentiation induction assay

Transfected and control cells were seeded on glass coverslips pre-coated with Poly-L-lysine in 24 wells plates at a density of $2 \times 10^4$ cells per well in DMEM 10% FBS 1% A/A. After 48 hours, media was replaced with DMEM/F12 supplemented with N2, B27 and GABA 100 µM (Differentiation medium) [22]. Cells were incubated at 37°C for 9 days, differentiation medium was replaced every day until cell fixation for further analysis.

## Experimental groups

Six groups were established. The first group was maintained under control conditions (CTRL), the second group was treated with 100 µM NMDA (NMDA), the third group received only the electroporation insult (MOCK), the fourth group was transfected with empty dCas9 vector (dCAS9), the fifth group was transfected with CRISPRi constructions for *Dnmt3a* knockdown (DNMT3a KD), and the sixth group was transfected with CRISPRi constructions for *Dnmt3a* knockdown with simultaneous stimulation with 100 µM NMDA (DNMT3a KD NMDA).

## Quantitative PCR

Total RNA was isolated from Müller cells cultures (previously treated as described below) using TRIzol reagent (Sigma-Aldrich) according to the manufacturer's instructions. The quality and quantity of isolated RNA were determined by spectrophotometric analysis (NanoDrop 2000, Thermo Scientific) and electrophoresis in 1% agarose gel. The cDNA was synthesized using 500 ng of RNA and the RevertAid First Strand cDNA Synthesis Kit (Thermo Fisher), following the manufacturer's instructions. cDNA samples were stored at −70 °C until further analysis. To assess the expression of some multipotency-associated genes and methylation-related genes, we employed SYBR green-based quantitative PCR (qPCR) that was performed on a Rotor Gene-Q system for qPCR (QIAGEN) to determine the expression of *Dnmt3a, Dnmt3b, Dnmt1, Ascl1, Lin28* and *Nestin* genes and *Gapdh* as housekeeping gene. The qPCR reaction included 1 µl of cDNA and 0.5 µM of specific primers for each gene (Table 3). Thermal cycler conditions were a three-step qPCR run with an initial denaturalization at 95°C for 10 minutes and then 40 cycles of 30 seconds at 95°C followed by 35 seconds at 63°C in the case of *Dnmt3a* and *Gapdh*, at 62°C in the case of *Dnmt3b* and *Dnmt1* or followed by 35 seconds at 60°C in the case of *Lin28, Ascl1* and *Nestin* and an extension step at 72°C for 40 seconds with fluorescence acquisition. All experiments were performed in triplicate, including the appropriate no-template control (NTC) and melt curve for each. The expression level of the analyzed genes was normalized against a housekeeper gene (GAPDH) determining any fold-change through the $2^{-\Delta\Delta Ct}$ method.

## Viability and cell proliferation

Muse Count and Viability assay kit was used to determine cell viability in each culture, before any other test, following the manufacturer's instructions. Briefly, we harvested treated cells with trypsin EDTA 0.12% solution in DMEM and centrifugated at 1200 x g for 5 min at room temperature. The supernatant was discarded, and the cellular pellet was resuspended in DMEM FBS 10% and A/A 1% at a cell density of ~$1\times10^6$ cells/mL. We mixed 50 µL of cell suspension and 450 µL of

**Table 3. List of qPCR primers.**

| Gene | Forward | Reverse |
|------|---------|---------|
| Dnmt3a | GCCGAATTGTGTCTTGGTGGATGACA | CCTGGTGGAATGCACTGCAGAAGGA |
| Dnmt3b | GCGCAGCGATCGGCGCCGGAGAT | CATACCCGCTGGCACCCTCTTCTTCA |
| Dnmt1 | CCTAGTTCCGTGGCTACGAGGAGAA | TCTCTCTCCTCTGCAGCCGACTCA |
| Lin28 | GGTCTGGAATCCATCCGTGTCA | TCCTTGGCATGATGGTCTAGCC |
| Ascl1 | GCAACCGGGTCAAGTTGGT | GTCGTTGGAGTAGTTGGGGG |
| Nestin | AGGAGAAGCAGGGTCTACAGAG | AGTTCTCAGCCTCCAGCAGAGT |
| Gapdh | ACTGGCATGGCCTTCCGTGTTCCTA | TCAGTGTAGCCCAAGATGCCCTTC |

Muse Count and Viability Reagent and placed in dark at room temperature for 5 min. Cells resuspended in assay buffer were analyzed on a Muse Cell Analyzer in accordance with the manufacturer's instructions.

Cell proliferation was assessed by detecting Ki67, a nuclear protein associated with cell cycle activity [43]. We used a Muse Ki67 Proliferation Kit following the manufacturer's instructions. Briefly, cells were starved from serum 24 h before the experiment. Harvested cells were washed with 1X PBS, centrifuged at $300 \times g$ for 5 minutes, and resuspended in 50 µL of 1X Fixation Buffer. The suspension was mixed thoroughly and incubated at room temperature for 15 minutes. After incubation, cells were washed once with 1X Assay Buffer, centrifuged at $300 \times g$ for 5 minutes, and resuspended in 100 µL of Permeabilization Buffer, followed by another 15-minute incubation at room temperature. Samples were then washed with 1X Assay Buffer, centrifuged again at $300 \times g$ for 5 minutes, and resuspended in 50 µL of Assay Buffer for an additional 15-minute incubation at room temperature. Subsequently, 10 µL of Muse Hu Ki67-PE reagent were added to each sample, mixed, and incubated for 30 minutes at room temperature in the dark. After staining, 150 µL of 1X Assay Buffer were added to each sample, and the samples were analyzed using the Muse Cell Analyzer. Data were expressed as the percentage of Ki67-positive cells.

## Immunocytochemistry

Müller cells were seeded on glass coverslips pre-coated with Poly-L-lysine placed in 24 wells plates with DMEM/10% FBS and 1%penicillin/streptomycin at a density of $2 \times 10^4$ cells per well. Cells were incubated at 37°C and 5% $CO_2$ for 2–7 days. Cells were rising in 1X PBS and then fixed in 4% paraformaldehyde (PFA) for 20 minutes at room temperature followed by two 1X PBS washes. Cell auto-fluorescence was quenched with a Tris-base (10 mM) glycine (100 mM) solution. The cells were permeabilized with 0.3% PBS-Triton and incubated in blocking solution for 2 hours at room temperature (5% Normal Goat Serum (NGS), 3% Bovine Serum Albumin (BSA) and 0.3% Triton). After that the samples were incubated with following primary antibodies: mouse monoclonal anti-βIII tubulin antibody (Santa Cruz sc-80005) (1:300); rabbit polyclonal anti-GS antibody (Abcam ab73593) (1:300); mouse monoclonal anti-nestin antibody (Santa Cruz sc-23927) (1:200); rabbit polyclonal anti-Lin28 (Abcam ab46020) (1:200) diluted in 1:3 blocking solution at 4°C overnight. Cells were rinsed five times with 0.01% PBS-Triton and then incubated with secondary antibodies goat anti-mouse IgG (Alexa 488) and donkey anti-rabbit IgG (Alexa 546) at a 1:800 dilution (Invitrogen) in 1:3 blocking solution, incubated for 2 hours at room temperature, and DAPI (Sigma-Aldrich D9542) was used for nuclei stain at a dilution of 1:500 in 1X PBS. Finally, coverslips were mounted in microscope slides, using ProLong ™ Diamond Antifade Mountant mounting medium (Invitrogen P36970), dried at room temperature for 2 hours and stored at 4°C. Fluorescent images were obtained using a ZEISS Axiovert 40 C/40 CLF inverted Fluorescent Microscope (Carl Zeiss, AXIO VISION Rel. 4.8 software) and an LSM 800 confocal system coupled to an inverted Axiovent AX10 microscope (Carl Zeiss, ZEN blue edition software). Non-stained samples were used as negative control. We performed at least three independent experiments per condition and evaluated ten non-overlapping fields per group; representative data showed at least 150 evaluated cells per group.

## Migration assay

The wound healing assay was performed as previously described [44] with minor modifications. Briefly, control and immediately transfected cells were seeded in 24 well-plates at a density of $5\times10^4$ per well and allowed to reach 90% confluence. Cells were starving with culture medium without serum 12 h before running the assay. A sterile SPLScar Scrather 24 well was used to create scratches across the center of the wells. After wounding, cells were rinsed with 1X PBS to eliminate cell debris, and the culture media was replaced with fresh medium. Cells were incubated at 37°C and 5% CO2 and "wound healing" capacity was evaluated at 0, 24 and 48 h. EGF was used as positive migration control [45].

Images of wound area were acquired in an inverted optical microscope (Leica) with a 10X objective, using an EC3 camera coupled to the microscope. The wound area was quantified using ImageJ software, measuring the distance between the edges of the wound at different time points. Migration rates were calculated as the percentage of wound closure relative to the initial wound area.

## DNMTs activity assay

Total DNMT enzymatic activity was quantified using DNMT Activity Quantification Kit (Abcam #ab113467) following the manufacturer's instructions. Briefly, cells were detached with 0.12% trypsin/EDTA solution (Sigma-Aldrich #T4049) and quenched by adding one volume of culture media with 10% FBS. Cells were centrifugated at 300 x g for 5 min and resuspended in 100 μL Pre-Extraction Buffer from Nuclear extraction kit (Abcam #ab113474) per $1\times10^6$ cells to obtain nuclear proteins. Cell suspension was placed in ice for 10 min, vigorously vortexed for 10 seconds and centrifuged for one minute at 12 000 rpm. Cytoplasmatic extract was discarded, and nuclear extract pellet was resuspended in about 20 μL of dithiothreitol (DTT) and protease inhibitor cocktail (PIC) solution (1:1000) and incubate in ice for 15 min with vortex every 3 min. An additional sonication was performed (3 times 10 s each). The resultant nuclear protein suspension was centrifuged at 14 000 rpm and the supernatant was transferred into a new clean tube. Protein concentration was measured by spectrophotometry (NanoDrop 2000, Thermo Scientific) and Bradford assay. About 10 μg of nuclear protein was used for each activity quantification. A 45–50 μL of AdoMet Working solution was added to each assay well and the mixture was incubated at 37°C for 120 min, tightly covered to avoid evaporation. After incubation, wells were washed with 150 μL of 1X Wash Buffer three times. 50 μL of capture antibody was added to each well, covered with Parafilm and incubated at room temperature for 60 min and then washed three times with 1X Wash Buffer. 50 μL of detection antibody was added to each well, covered with parafilm and incubated at room temperature for 30 min followed by four washes with 1X Wash Buffer. 50 μL of Enhancer Solution was added to each well, covered with Parafilm and incubated at room temperature for 30 min and washed five times, then 100 μL of Developer Solution was added to each well and incubated at room temperature for 1–10 minutes protected from light followed by the addition of 100 μL of Stop Solution. Absorbance was read on a microplate reader (Sunrise™ absorbance reader TECAN) at 450 nm wavelength. Activity was reported as optical density per hour of incubation per protein concentration (OD/h/μg). Three independent experiments were conducted for each group and presented as normalized data versus control (CTRL).

## Results

### DNMT3a knockdown is sufficient to promote Müller glia dedifferentiation

To investigate the role of *Dnmt3a* in MG dedifferentiation, we first conducted a time-course analysis of *Dnmt3a* expression in response to NMDA (100 μM) exposure. RT-qPCR analysis revealed a transitory increase of *Dnmt3a* expression (~20-fold relative to control, $p<0.05$), initiating at 4 hours after treatment and peaking at 24 h of continuous stimulation and declined to baseline after 48 h (Fig 1A). Next, we designed single guide RNAs (sgRNAs) targeting the promoter regions of the *Dnmt3a* gene. These sgRNAs were incorporated into CRISPR interference systems (CRISPRi) utilizing a catalytically inactive Cas9 (dCas9) fused to the Krüppel-associated box (KRAB) repressor domain [46] (S1). This configuration

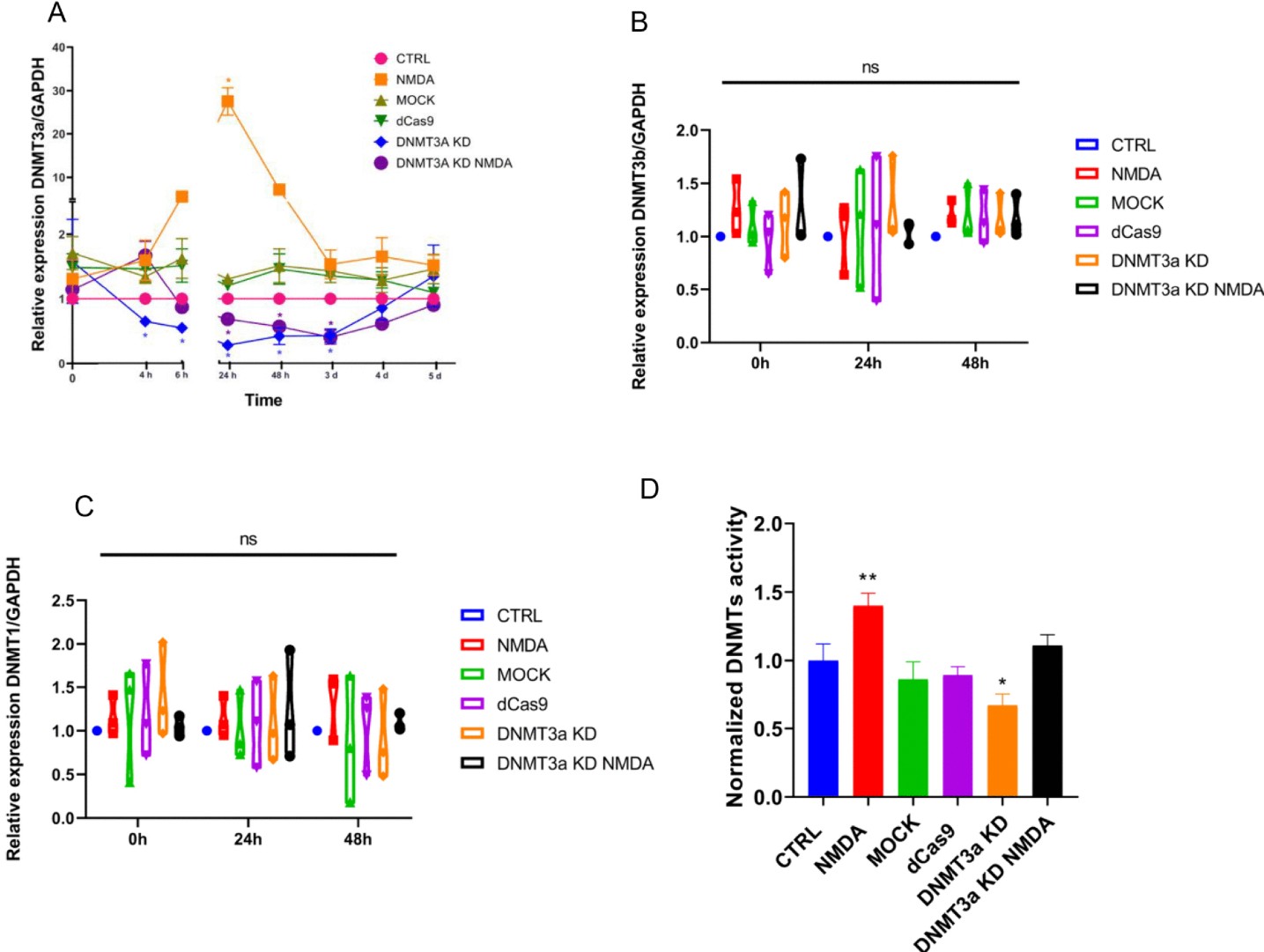

**Fig 1. DNMT3a knockdown system alters DNMT expression and activity in rodent MG cell cultures.** (A) Time-course analysis of *Dnmt3a* mRNA expression normalized to GAPDH, measured by qRT-PCR. A transient *Dnmt3a expression* increase was observed at 24 h following NMDA treatment. DNMT3a KD and DNMT3a KD NMDA groups showed sustained suppression of *Dnmt3a* expression initiating at 24 h after transfection and finishing at 4 days after transfection. Data are shown as mean ± SEM; *$p < 0.05$ versus CTRL at the corresponding time point (two-way ANOVA with Tukey's post hoc test). (B–C) Relative expression of *Dnmt3b* and *Dnmt1* mRNA normalized to GAPDH at 0 h, 24 h, and 48 h after CRISPRi plasmid transfection showed no significant differences among groups (ns, not significant; one-way ANOVA). (D) Global DNMT enzymatic activity normalized to CTRL. NMDA significantly increased DNMT activity (**$p < 0.01$), while DNMT3a KD significantly reduced it (*$p < 0.05$). Activity was restored in the DNMT3a KD NMDA group. CTRL: control, NMDA: exposure to NMDA 100 μM, MOCK: electroporation without plasmid, dCas9: transfection of empty plasmid, CRISPRi: CRISPR interference-mediated *Dnmt3a* knockdown, CRISPRi NMDA: *Dnmt3a* knockdown with simultaneous exposure to NMDA 100 μM. Data represents SEM from three independent experiments (one-way ANOVA with Tukey's post hoc test).

enables targeted transient transcriptional repression of *Dnmt3a* expression in transfected cells. To assess the efficiency of the silencing system, we compared expression in MG cultures exposed to NMDA (100 μM), transfected with the *Dnmt3a* knockdown plasmid (DNMT3a KD), or treated with both interventions (DNMT3a KD NMDA). Three control groups were included to ensure that the observed effects were not attributable to culture conditions (CTRL), the transfection procedure (MOCK), or the presence of the empty plasmid (dCas9).

We confirmed that CRISPRi treatment (DNMT3a KD) efficiently downregulated *Dnmt3a* expression (~0.5 fold relative to control, p < 0.05), with reduced expression maintained for up to 3 days and maximal inhibition 24 h after transfection (Fig 1A and S2 Fig). DNMT3a KD and NMDA simultaneous treatments did not reach the expression levels observed with NMDA alone and instead showed a response similar to DNMT3a KD alone, reaching maximal suppression at 3 days, followed by a full return to baseline levels by day 5 (*p* < 0.05 vs. control). Across all expression analyses, no significant differences were observed among the control groups (CTRL, MOCK, and dCas9). Specificity of the effect of the inhibition system on *Dnmt3a* expression was confirmed by RT-qPCR evaluation of the expression of *Dnmt3b* and *Dnmt1* under the same experimental conditions. No significant effects were observed (Fig 1 B, C), although possible effects of *Dnmt3b* and *Dnmt1* were not evaluated beyond qPCR.

To investigate whether the transcriptional downregulation of *Dnmt3a* was associated to alterations in DNMT enzymatic activity we isolated nuclear protein extracts from Müller glial cells and performed an immunoassay (Fig 1D). NMDA exposure significantly increased methylation activity (p < 0.01), while *Dnmt3a* knockdown significantly reduced it (p < 0.05). Remarkably, methylation in the DNMT3a KD NMDA group was comparable to control levels, indicating that *Dnmt3a* knockdown can revert NMDA-induced DNMT activation. These results support a role for *Dnmt3a* in Müller glia dedifferentiation capacity (Fig 1D).

Next, we analyzed the effect of *Dnmt3a* knockdown on the capacity of MG cultures to induce the expression of retinal progenitor cells-related markers: *Ascl1* (Fig 2A), *Nestin* (Fig 2B) and *Lin28* (Fig 2C). NMDA exposed MG cells exhibited transitory increases in *Lin28*, *Ascl1* and *Nestin* expression that peaked at 24 h (*Lin28* and *Ascl1*, ~4-fold vs. control, p < 0.05) or 6 h (*Nestin*, ~20-fold vs. control, p < 0.05) that rapidly decreased to reach basal levels by 48 h. In contrast, in the DNMT3a KD group, both *Lin28* and *Ascl1* expression levels were markedly elevated (~6-fold compared to control, p < 0.05) and sustained for a longer time (up to 3 days) compared to control and NMDA stimulation. *Nestin* expression did not reach the levels observed after NMDA exposure, however, it exhibited a sustained expression (~10-fold vs. control, p < 0.05) over four days, contrasting with transient peak induced by NMDA at short times. The combination of DNMT3a KD with NMDA stimulation resulted in the highest induction of *Lin28* (~7-fold vs. control groups, p < 0.05) and *Ascl1* (~8-fold vs control, p < 0.05). *Nestin* exhibited a similar expression pattern to that observed in the DNMT3a KD group, with expression increasing from 6 hours post-transfection, peaking at 48 hours, and subsequently returning to baseline levels by day 5 (Fig 2B and S2 Fig). Immunofluorescence analysis and fluorescence quantification confirmed that Lin28 protein expression is significantly (p < 0.05) induced in DNMT3a KD samples (Fig 2D, E).

These results suggest that *Dnmt3a* may play a repressive role in the expression of retinal progenitor cell markers.

Changes in *Dnmt3a* expression were accompanied by morphological changes in Müller cells. Quantification of cell area demonstrated that cells exposed to NMDA presented a significant increase compared to control groups (CTRL, MOCK and dCas9) (Fig 3A). In contrast, *Dnmt3a* knockdown (KD) cells, whether treated with NMDA or not, displayed a reduced cell area relative to the controls (Fig 3A). These findings suggest that epigenetic modifications induced by *Dnmt3a* knockdown may enhance other responses in Müller glia, potentially related to the acquisition of stem cell-like properties.

## Proliferation and cell migration are upregulated during DNMT3a knockdown in MG cultures

In some vertebrates, after retinal damage, dedifferentiated MG could proliferate as part of stem cell-like phenotype acquisition [47], and migrate to injured sites to participate in retinal regeneration by differentiating to replace lost cells [48,49]. To characterize the proliferative response of Müller cells, we quantified the percentage of cells immunopositive to Ki67 (Fig 3B). A basal level of Müller cell proliferation was observed in the control groups (CTRL, MOCK, dCas9), with approximately 10% of cells positive for Ki67. In the NMDA-treated group, the proportion of Ki67-positive cells increased to about 30%, indicating re-entry into the cell cycle. In contrast, *Dnmt3a* knockdown groups, both with and without NMDA treatment, showed approximately 40% Ki67-positive cells, suggesting that *Dnmt3a* knockdown alone enhances cell

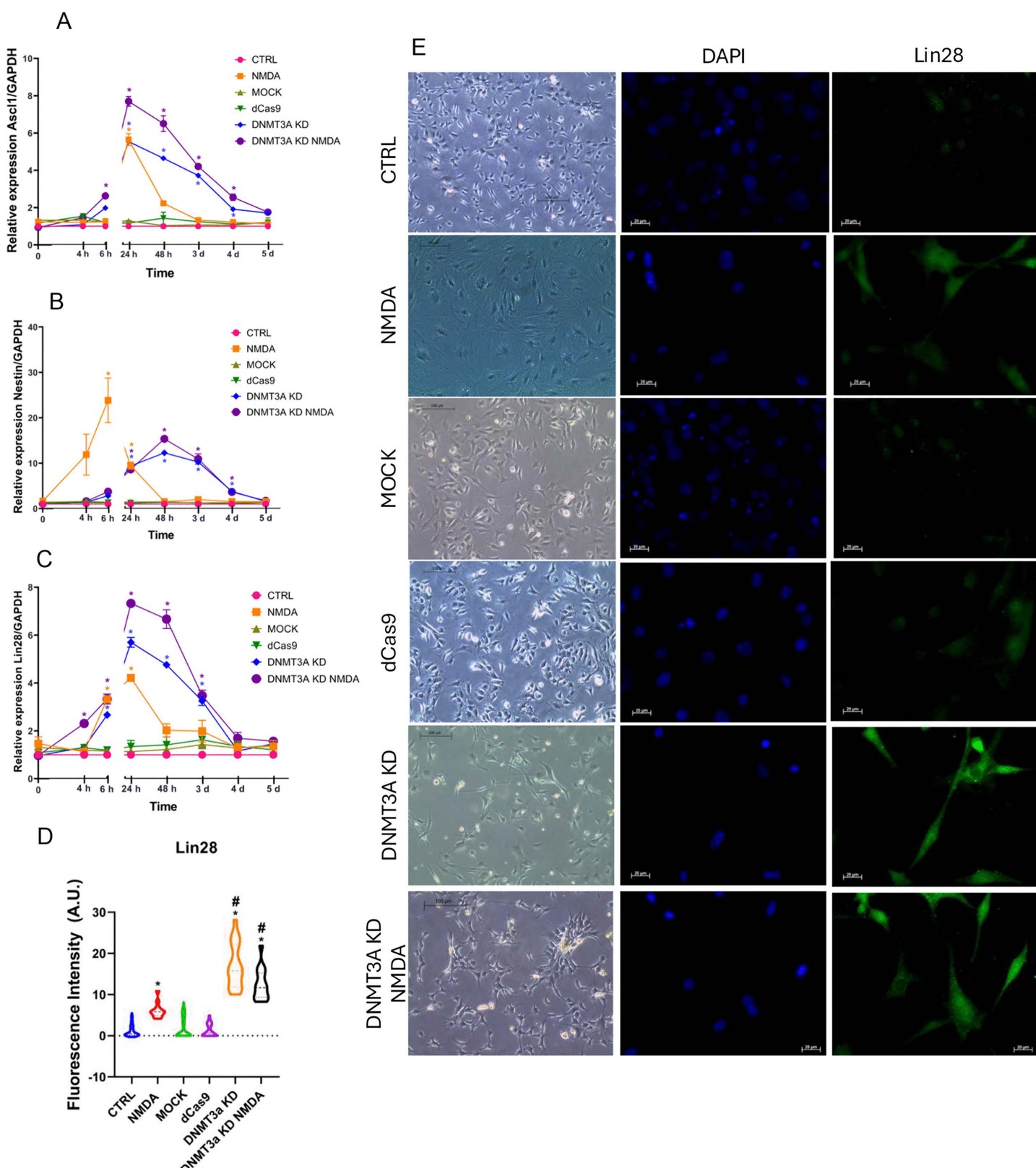

**Fig 2. DNMT3a knockdown promotes expression of progenitor markers in MG cell cultures.** (A–C) Time-course analysis of mRNA expression levels of progenitor markers *Ascl1, Nestin* and *Lin28*, normalized to *Gapdh*, measured by qRT-PCR. NMDA and DNMT3a KD alone transiently upregulated

these markers, with a more pronounced and sustained effect observed in the DNMT3a KD NMDA group. Data represent mean ± SEM of three independent experiments; *p < 0.05 vs. CTRL at the corresponding point (two-way ANOVA with Tukey's post hoc test). (D) Quantification of Lin28 fluorescence intensity in arbitrary units (A.U.) from immunocytochemistry images shown in (E). Significant increases in Lin28 expression were observed in NMDA, DNMT3a KD and DNMT3a KD NMDA groups compared to CTRL (*p < 0.05 vs. CTRL; #p < 0.05 vs. NMDA; one-way ANOVA with Tukey's post hoc test). (E) Representative phase contrast and immunofluorescence images showing DAPI (nuclei, blue) and Lin28 (green) staining at 48 h post-treatment. CTRL: control, NMDA: exposure to NMDA 100 μM, MOCK: electroporation without plasmid, dCas9: transfection of empty plasmid, CRISPRi: CRISPR interference-mediated *Dnmt3a* knockdown, CRISPRi NMDA: *Dnmt3a* knockdown with simultaneous exposure to NMDA 100 μM. Phase contrast images scale bar = 100 μm, 10X; Immunofluorescence images scale bar = 20 μm, 40X.

proliferation, and this effect is not further influenced by NMDA exposure (Figure 3C). None of the treatments led to a reduction in cell viability (Fig 3D).

These findings suggest that epigenetic modulation by *Dnmt3a* knockdown promotes MG reentry to cell cycle.

Given the critical role of Müller glia (MG) migration in retinal regeneration, we established a wound healing assay to assess whether cellular migration is affected under *Dnmt3a* knockdown conditions. A mechanical scratch was introduced into confluent cell cultures for each experimental group (Fig 4A), and the scratch width was measured at 24 and 48 hours post-injury. NMDA exposure, *Dnmt3a* knockdown, and their combination (DNMT3a KD NMDA) enhanced Müller glia migration into the wound area, with migration initiating by 24 hours and complete wound closure observed by 48 hours. Remarkably, both DNMT3a KD and DNMT3a KD + NMDA-treated cells exhibited approximately 90% wound closure at 24 hours (Fig 4B), suggesting that *Dnmt3a* knockdown accelerates the migratory response of Müller cells independently of NMDA stimulation. In control groups (CTRL, MOCK and dCas9) complete closure was not achieved at any time point, and no significant differences were identified among these groups (Fig 4B).

### Transitory DNMT3a knockdown enhances pro-neural differentiation of dedifferentiated MG

MG cell identity and differentiation towards a neuronal-like phenotype was evaluated by immunofluorescence analysis using antibodies against glutamine synthetase (GS), a glial cell marker, and βIII-tubulin, an early neuronal marker (Fig 5). Both the number of immunopositive cells and the fluorescence intensity for each marker were quantified in control or transfected cells (in the absence or presence of NMDA (100 μM)) 48 hours after procedure as indicated in Fig 5A.

GS (red) was consistently expressed in over 95% of the analyzed cells (Fig 5B,C), suggesting that glial identity was preserved across all experimental conditions. As expected, βIII-tubulin (green) was undetectable in control groups (CTRL, Mock and dCas9). It was also undetectable in NMDA treated cells, suggesting that NMDA is not sufficient to induce neuronal differentiation. In contrast, DNMT3a KD groups presented significant (p < 0.01) increases of approximately 20% in the number of βIII-tubulin immunopositive cells, and in the fluorescence intensity for this marker (Fig 5 B-E). These results suggest that *Dnmt3a* knockdown could generate a permissive epigenetic landscape to promote the expression of several genes, including neuronal-like markers.

Based on previous observations that the neurogenic potential of rodent MG can be induced by exposure to a neuronal differentiation medium that incorporates GABA (100 μM) [22], we decided to evaluate the effect of DNMT3a KD in GABA-exposed MG cultures (Fig 6). Cells were exposed to a neuronal differentiation medium containing GABA for 9 days, as shown in Fig 6A. The NMDA groups (transfected or untransfected) were pre-treated with 100 μM NMDA to induce dedifferentiation prior to GABA exposure.

Under these conditions, we observed that while GS (red) expression is maintained across the experimental groups, NMDA induces a moderate increase in the number of βIII-tubulin (green) of about 20% (p < 0.05) and a more robust increase in DNMT3a KD (p < 0.001) and DNMT3a KD NMDA groups (p < 0.01) with percentages between 60–80% approximately (Fig 6B-E). The photomicrographs in Fig 6B reveal morphological changes consistent with neuronal-like differentiation. *Dnmt3a* knockdown cells exhibited an elongated shape with extended projections resembling neuronal processes, in contrast to the radial morphology observed in control Müller glial cells. However, interestingly we also observed a

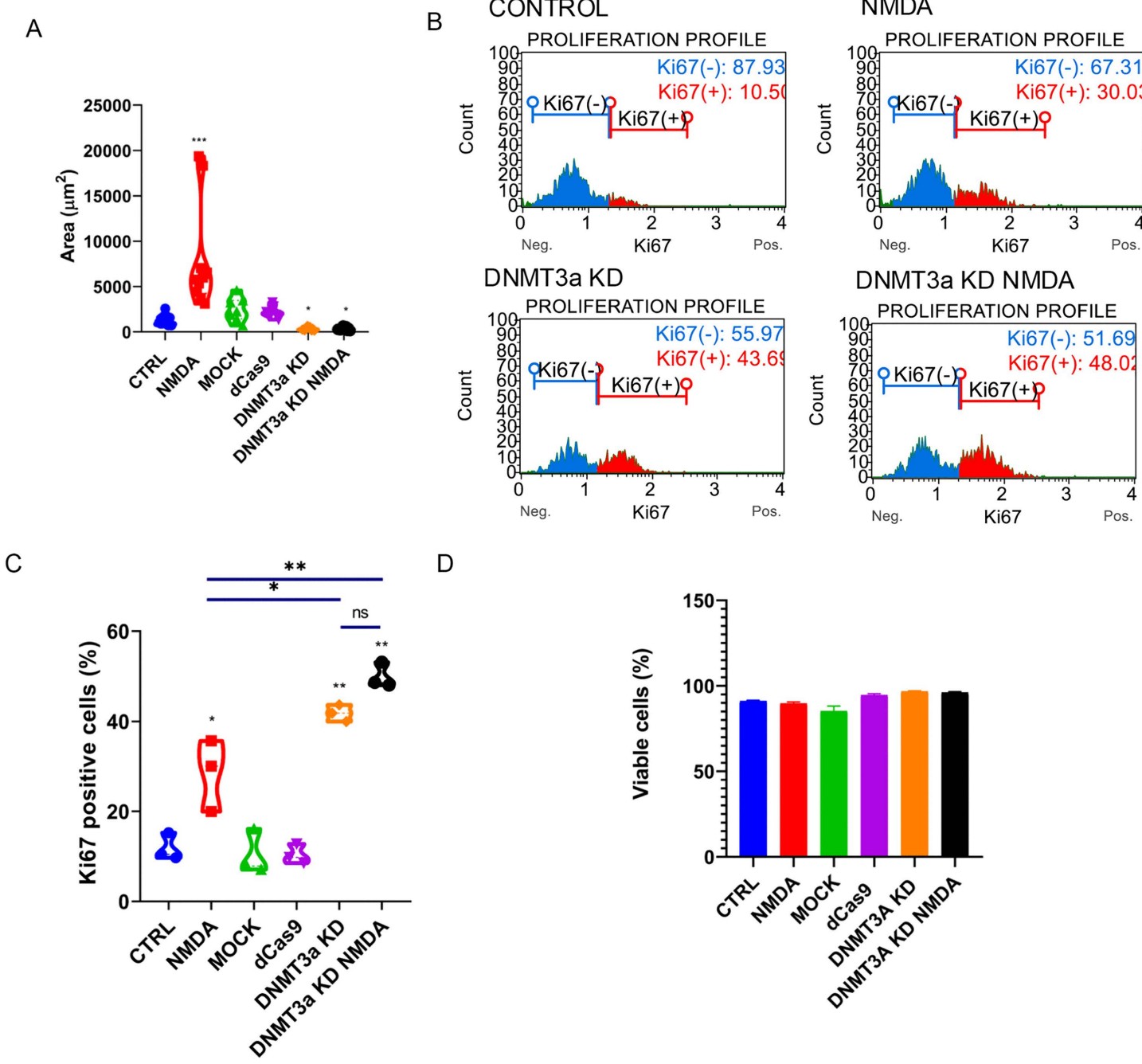

**Fig 3. DNMT3a knockdown enhances MG proliferation.** (A) Quantification of cell area (µm²) in control conditions (CTRL), with 100 µM NMDA exposition (NMDA), in electroporation only group (MOCK), in plasmid transfection control (dCas9), cells transfected with CRISPRi plasmid for *Dnmt3a* knockdown (DNMT3a KD) and CRISPRi transfected cells with simultaneous exposition to 100 µM NMDA (DNMT3a KD NMDA). Data are presented as violin plots *p < 0.05, ***p < 0.001 vs. CTRL. (B) Representative flow cytometry histograms for Ki67 detection across experimental conditions where NMDA, DNMT3a KD and DNMT3a KD NMDA show an increase in Ki67 positive cells compared to control. (C) Quantification of Ki67 + cells by flow cytometry confirms an enhancement in cell proliferation after NMDA exposition, while DNMT3a KD is sufficient to induce a higher fraction of proliferating cells and is not significantly increased by NMDA simultaneous stimulation; *p < 0.05, **p < 0.001, ns = no significant (one way ANOVA with Tukey´s post-hoc test. (D) Cell viability shows no significant differences among conditions.

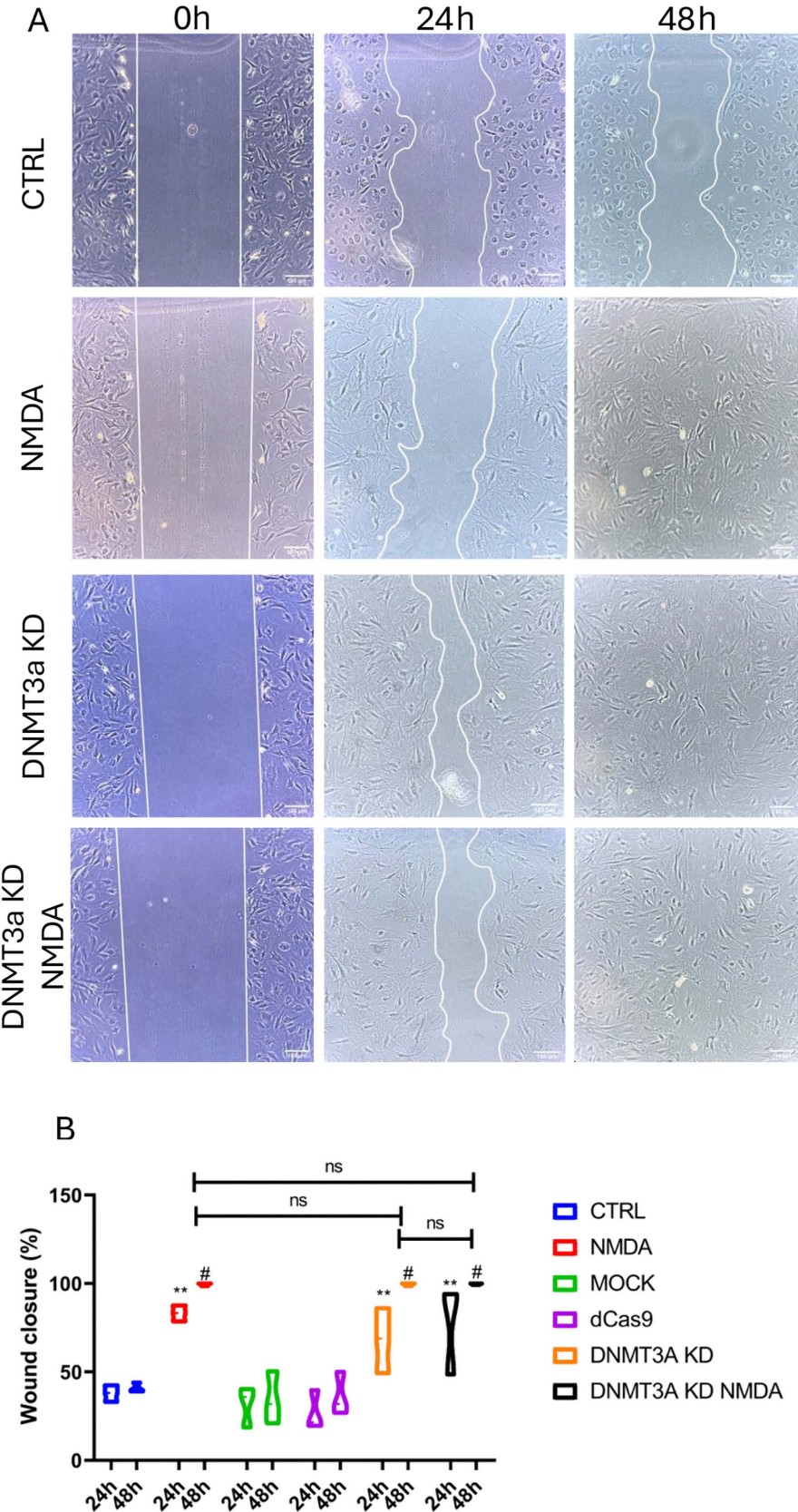

**Fig 4. DNMT3a knockdown and NMDA treatment enhance MG cell migration.** NMDA and *Dnmt3a* knockdown promote Müller glia migration in a wound healing assay. (A) Representative figures of Müller glia wound healing assay performed under six different conditions: CTRL: control, NMDA: exposure to NMDA 100 µM, MOCK: electroporation without plasmid, dCas9: transfection of empty plasmid, DNMT3a KD: CRISPR interference-mediated *Dnmt3a* knockdown, DNMT3a KD NMDA: DNMT3a knockdown with simultaneous exposure to NMDA 100 µM. Cultures were evaluated at initial time (0h), 24 h and 48 h. (B) Wound closure percentage was calculated from at least three independent experiments, data are shown as mean ± SEM, (one-way ANOVA with Tukey's post hoc test were performed *$p < 0.05$, **$p < 0.01$, ***$p < 0.001$, ns = no significant).

consistent co-expression of neuronal and glial markers (Fig 7). Together our data would suggest that *Dnmt3a* knockdown facilitates partial reprogramming of MG by neuronal-like differentiation allowing acquisition of neuronal features but with no losing of glial specific markers.

## Discussion

In this study we demonstrate that *Dnmt3a* knockdown unlocks the dedifferentiation and neurogenic potential in mouse MG. Using a CRISPR interference (CRISPRi) system, we demonstrate that *Dnmt3a* modulation alone can recapitulate key features of the regenerative responses observed in lower vertebrates such as zebrafish [50,51]. Dnmt3a KD and an excitotoxic stimuli, such as NMDA 100 µM, can synergize to amplify MG reprogramming potential.

MG exhibit distinct activation patterns in response to retinal damage, which vary across vertebrate species [52]. Activation can lead to a regenerative or to a non-regenerative response. For example, in regenerative species like zebrafish or young chicks, MG can initiate a response characterized by re-entry to cell cycle and activation of signaling pathways such as MAPK [53] which facilitates the transition from a differentiated state to a dedifferentiated and remains upregulated for 3–7 days after damage [23]. In vertebrates like mammals, MG activation is characterized by upregulation of gliotic genes and activation of signaling pathways related to immune response [52,54], and to a lesser extent to a transient expression of cell cycle regulators and neurogenic factors, but these rapidly return to quiescence, limiting the regenerative window [52,55]. Those gene expression alterations may be modulated by epigenetic changes, that also vary widely among different species [56,57].

We have previously demonstrated that an initial wave of DNA demethylation is essential for preserving the dedifferentiation capacity of mammalian Müller glia in response to high concentrations (100 µM) of NMDA [27]. This early event occurs within the first 6 hours of NMDA exposure, and it is associated to an increased expression of the Ten-Eleven Translocation 3 (TET3) DNA demethylase as demonstrated by loss-of-function experimental approaches [27]. We observed here that NMDA exposure also induced a subsequent transient and specific increase in *Dnmt3a* expression, peaking at 24 h. We could speculate that an increase of a DNA methyl transferase activity, could repress the expression of progenitor-associated genes, restricting the duration and magnitude of the regenerative response. To test this hypothesis, we performed a *Dnmt3a* knockdown via CRISPRi that effectively suppresses this upregulation and maintains low expression levels for several days, indicating that this approach is both effective and temporally stable. *Dnmt3a* knockdown alone led to a sustained upregulation of key genes in MG dedifferentiation such as *Lin28* and *Ascl1* and genes related to neural progenitor cells, such as *Nestin*. These genes are part of a particular axis indispensable for MG reprogramming [12,58–60]. This extended expression pattern resembles what has been documented in regenerative organisms such as zebrafish [1,7,47], where persistent *Lin28* and *Ascl1* expression is a hallmark of successful MG reprogramming. NMDA and *Dnmt3a* KD combination yielded synergistic effects on progenitor-related genes expression. These results support the hypothesis that *Dnmt3a* may act downstream or in parallel to stress signals to enforce a regenerative or non-regenerative state in Müller glia cells.

Beyond transcriptional changes, *Dnmt3a* knockdown also induced differential functional features, hallmarks of acquisition of a dedifferentiated state. These included cell proliferation by re-entry to cell cycle, enhanced migratory behavior and morphological changes. MG proliferation and migration are particularly necessary during retinal regeneration

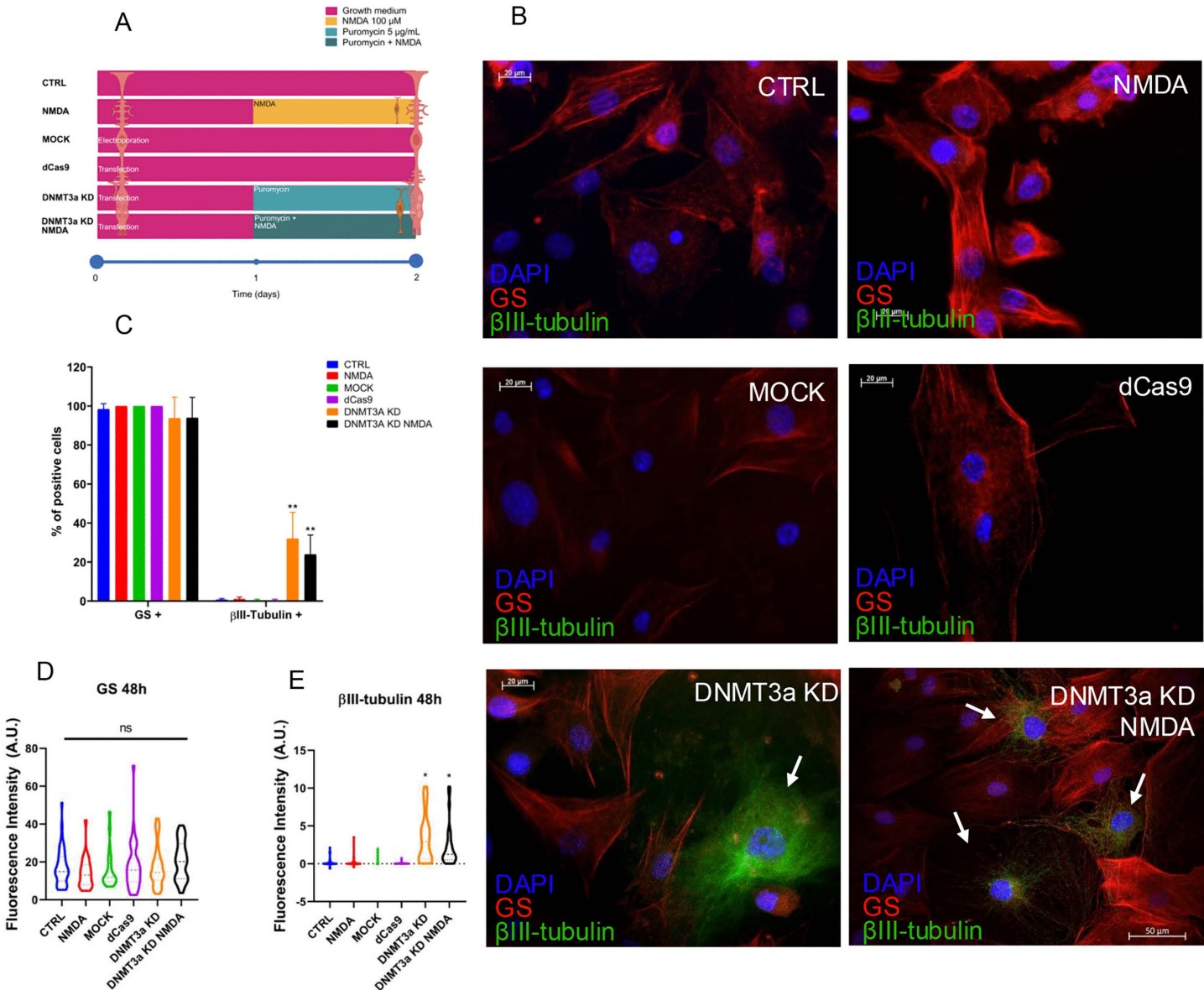

**Fig 5. DNMT3a knockdown induces expression of the early-neuron marker βIII-tubulin marker in MG cultures.** (A) Experimental design for early-neuron marker βIII-tubulin detection, time course for sample acquisition and chemical treatment. (B) Representative immunofluorescences of MG at 48 h after treatments, cultured in basal growth media, CTRL: control, NMDA: exposure to NMDA 100 µM, MOCK: electroporation without plasmid, dCas9: transfection of empty plasmid, DNMT3a KD: CRISPR interference-mediated *Dnmt3a* knockdown, DNMT3a KD NMDA: *Dnmt3a* knockdown with simultaneous exposure to NMDA 100 µM; show GS (red), βIII-tubulin (green), and nuclei (DAPI, blue). (C) Percentage of positive cells to GS and βIII-tubulin, where all conditions present constant expression of GS, while βIII-tubulin percentage of positive cells were increased in DNMT3a KD and DNMT3a KD NMDA. (D) Quantification of fluorescence intensity of GS marks at 48 h after treatment, no significant differences were observed across conditions. (E) Quantification of fluorescence intensity of βIII-tubulin shows an increase in DNMT3a KD and DNMT3a KD NMDA groups, consistent with the increase in βIII-tubulin percentage of positive cells. Scale bars = 20 µm, 40X. Percentage of positive cells for each marker were presented as bar graphic for mean ± SEM from three independent experiments, counting a minimum of 10 non-overlapped fields and at least 150 cells per group. *p < 0.05, **p < 0.01, ns = no significant (one-way ANOVA with Tukey post hoc test).

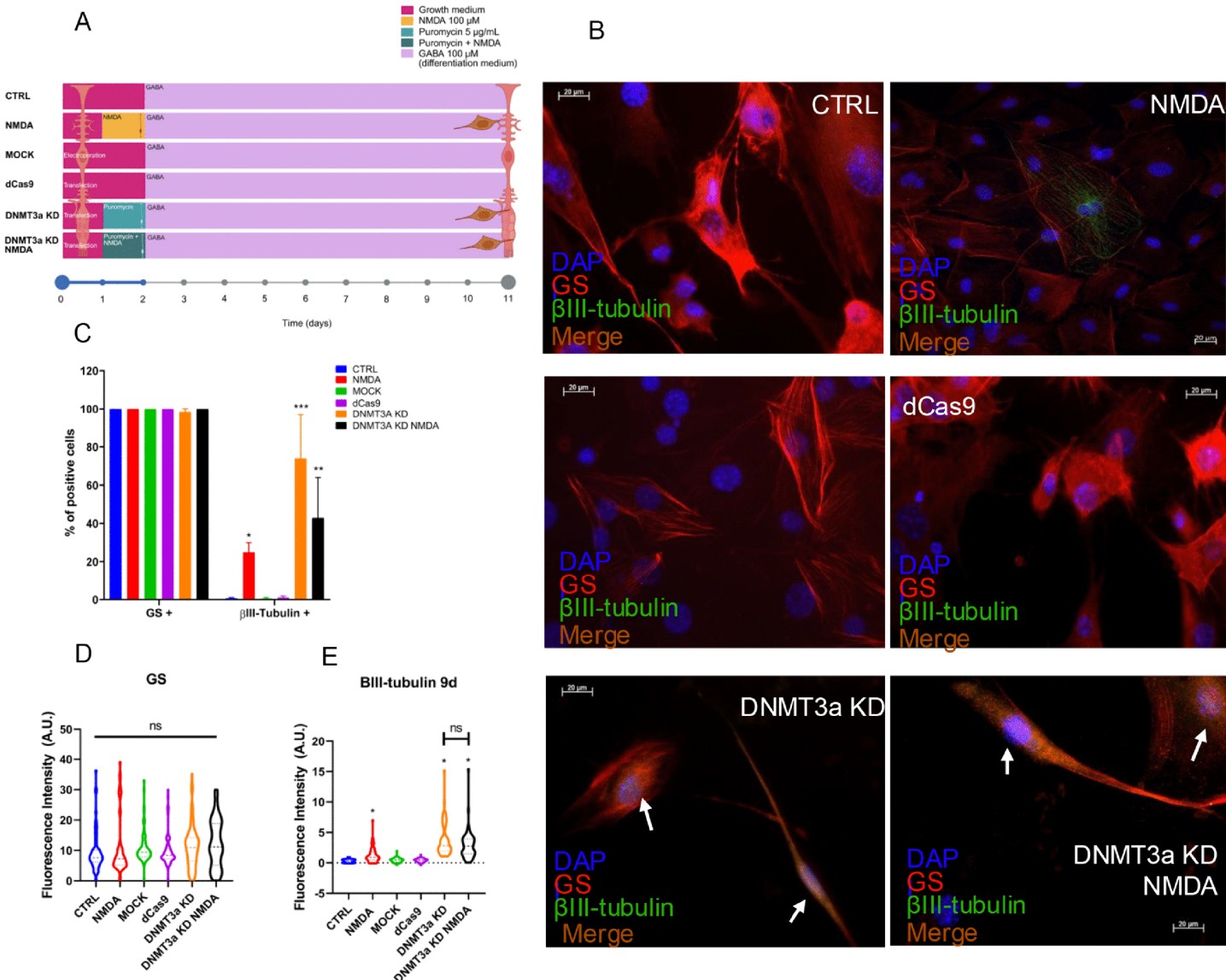

**Fig 6. DNMT3a knockdown enhances the expression of early neuronal-marker βIII-tubulin after GABA treatment.** (A) Experimental design for early-neuron marker βIII-tubulin induction with GABA after CRISPRi transfection, showing the time course followed for sample acquisition and chemical treatment. (B) Representative images for immunofluorescence of MG cultured 48 h in growth medium and then cultured for 9 days in differentiation medium with GABA 100 μM. Images show GS (red), βIII-tubulin (green), and nuclei (DAPI, blue). Control conditions (CTRL, MOCK, dCas9) maintain GS expression with negligible βIII-tubulin signal. Meanwhile in NMDA, DNMT3a KD and DNMT3a KD NMDA βIII-tubulin positive cells were present, and GS expression was maintained. CTRL: control, NMDA: exposure to NMDA 100 μM, MOCK: electroporation without plasmid, dCas9: transfection of empty plasmid, DNMT3a KD: CRISPR interference-mediated *Dnmt3a* knockdown, DNMT3a KD NMDA: *Dnmt3a* knockdown with simultaneous exposure to NMDA 100 μM. Scale bars = 20 μm, 40X. (C) Quantification of percentage of positive cells to each marker, all conditions present constant expression of GS, while βIII-tubulin percentage of positive cells were increased in NMDA, DNMT3a KD and DNMT3a KD NMDA. (D) Quantification of fluorescence intensity of GS marks at 48 h after treatment, no significant differences were observed across conditions. (E) Quantification of fluorescence intensity of βIII-tubulin shows an increase in NMDA, DNMT3a KD and DNMT3a KD NMDA groups. Data were presented as violin plots of at least three independent experiments, counting a minimum of 10 non-overlapped fields and at least 150 cells per group. *$p < 0.05$, ***$p < 0.001$, ns = no significant (one-way ANOVA with Tukey's post hoc test).

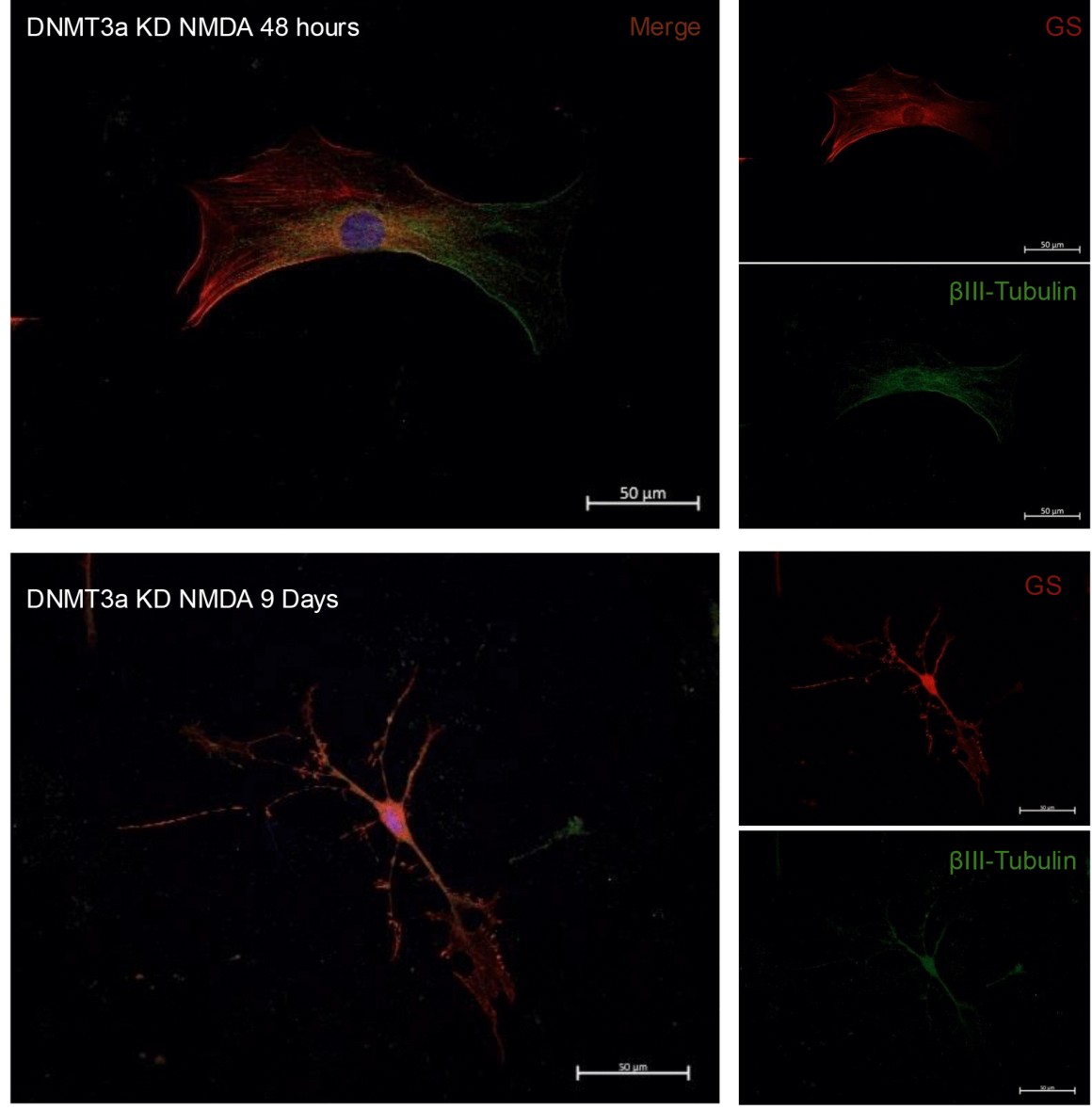

**Fig 7. Neuronal and glial gene marker co-expression in DNMT3a KD +NMDA cells.** Representative immunofluorescences of MG at 48 h and 9 days after CRISPRi transfection with simultaneous addition of NMDA 100 μM, where we could observe coexpresión of glial (GS, red) and early-neuronal (βIII-tubulin, green) markers indicating a partial cell reprogramming. DNMT3a KD NMDA: *Dnmt3a* knockdown with simultaneous exposure to NMDA 100 μM; show GS (red), βIII-tubulin (green), and nuclei (DAPI, blue). Scale bars = 50 μm, 40X.

[30,47–49,61]. In some cells, even in pathological states, deletion of *Dnmt3a* is related to an increase in these processes [62,63]. This is consistent with our findings and suggests that the resulted hypomethylation state affects not only the expression of progenitor-related markers, but also the expression of key signaling pathways regulators involve in cell survival, proliferation and migration [38,40,64–66].

Furthermore, we demonstrate that *Dnmt3a* knockdown (KD) Müller glia (MG) express βIII-tubulin, a marker of early neuronal differentiation, 48 hours after transfection; an expression not observed during NMDA-induced activation of MG in the absence of additional differentiation signals. In the presence of the neurogenic signal GABA [22], DNMT3a KD cells exhibit

further increases in βIII-tubulin expression and adopt a neuronal-like morphology. Interestingly, we find that both at early and later time points, 60–80% of the cells co-express glial and neuronal markers. Co-expression of neuronal and progenitor markers has been previously reported in mature mouse MG induced to undergo neurogenic differentiation via viral-mediated Ascl-1 overexpression [12]. This sustained expression of progenitor markers was associated to a possible limitation in the full differentiation capacity of MG under those experimental conditions. Whether we are observing a similar partial reprogramming process, and whether this may affect functionality of the cells expressing neuronal markers, remain to be further investigated. Further genotypic characterization of βIII-tubulin–expressing cells, through the evaluation of their immunoreactivity to MAP2 or NeuroD1 antibodies, as well as electrophysiological assessment of their intrinsic properties, is required to confirm their neuronal functionality. Such analyses could also help determine whether, for instance, temporal control of DNMT3a expression might extend the window of neuronal differentiation, thereby leading not only to an increased number of neurons but also to the acquisition of a fully functional neuronal phenotype. Our findings demonstrate that *Dnmt3a* is a key repressor for MG reprogramming, that may serve as an epigenetic barrier reversing an initial wave of NMDA-induced DNA demethylation. *Dnmt3a* CRISPR-dCas9-mediated downregulation may represent a promising strategy to prolong the transient progenitor state of MG and enhance their neurogenic potential and retinal regenerative capacity in mammals. Importantly, this approach could complement several reprogramming strategies previously applied in both regenerative and non-regenerative species (Table 4).

Orthogonal validation methods, such as small-molecule epigenetic modulators [72], siRNAs [27], or the recently characterized and optimized CRISPR protein family Cas13, also known as CRISPR-CasRx [71], would be invaluable to reinforce the robustness and significance of these findings.

**Table 4. MG reprogramming strategies.**

| Reprog. Strategy | Zebrafish | Mammals | Efficiency | References |
|---|---|---|---|---|
| Injury-induced | Induction of: Ascl1a, lin28a MG proliferation MGDP generation Neurogenesis | Induction of: Reactive gliosis Insufficient neurogenic response | **Zebrafish: High** *Permisive epigenetic landscape* **Mammals: Low** *Non-permisive epigenetic landscape (DNA methylation block)* | Zebrafish: [7,59], Mammals: [2,11,26] |
| Proneural TF over-expression (ASCL1/ Ascl1a) | Ascl1a endogenously required. Forced Ascl1a expression increases reprogramming. | Ascl1 overexpression in mouse induces early stages of retinal regeneration) | **Zebrafish: effective** **Mouse: moderate** if combined with other interventions (age/ injury/epigenetic). | Zebrafish: [59] Mammals: [13,14] |
| Epigenetic modula-tion HDAC inhibitors | HDAC inhibition reversibly blocks regeneration | HDACi synergize with ASCL1 to permit repro-gramming of murine MG | **Zebrafish:** histone acetylation changes required. **Mouse:** HDACi+Ascl1 increases efficiency | Zebrafish: [67] Mammals: [14] |
| DNA methylation modulation | DNMT inhibition perturbs MG reprogramming<br><br>DNA demethylation inhibiton attenuates retinal regeneration | DNMT pharmacological inhibition prevents MG dedifferentiation<br><br>Tet3-mediated DNA demethylation is essen-tial for maintaining MG dedifferentiation capacity | **Zebrafish:** characterized by a dynamic DNA methylation landscape.<br><br>**Mouse:** Only partial. Increased pluritpotency gene expression and induction of initial stages of neurogenesis | Zebrafish: [23,68,69]<br><br>Mice: [26,27]<br><br>Chick: [24] |
| CRISPR-dCas9/ CasRx (Cas13d) based approaches | Tools used to determine function of specific reg-ulators and cell-specific responses | Emerging strategy: offers precision vs. pharmacological modulation | **Zebrafish and mammals:** requires optimization (delivery, efficiency, off-targeting, safety for in vivo procedures, durability of the effects) | Zebrafish: [38,70] Mouse: [71], this work |

For these observations to have a physiological impact, it is essential to translate these studies into an *in vivo* system, where the interaction of Müller cells with all retinal and immune cells, including microglia, can be properly taken into account. Although complex, this is a feasible task. The CRISPR/Cas system has been successfully adapted to target retinal cells *in vivo*, either through electroporation [73,74] or by adeno-associated virus 2 (AAV2)-mediated deliver, an approach that holds greater promise for future clinical applications [75]. CRISPRi systems offer high efficiency and specificity, when compared for example with the available non-specific pharmacological DNMT inhibitors (e.g., 5-azacytidine or RG108) which broadly reduce methylation across the genome and may indirectly affect multiple cellular pathways, often leading to cytotoxicity or limited temporal control. The use of a catalytically inactive Cas9 (dCas9) fused to the KRAB repressor domain enabled precise, locus-specific and transient transcriptional repression of Dnmt3a, simultaneously minimizing off-target disturbances and preserving the necessary temporal reversibility of the epigenetic alteration. In addition, these procedures are scalable and should permit the evaluation of the coordinated regulation of other transcription factors.

However, CRISPR/Cas9 gene editing technologies are not without limitations, and resolving such limitations could overcome some of the potential risks that partial reprogramming of Müller glia carry. For example, incomplete or unstable reprogramming may lead to cellular heterogeneity, with subpopulations that co-express glial and neuronal markers but fail to achieve full neuronal functionality. Such intermediate states could result in aberrant signaling, impaired glial support functions, or even uncontrolled proliferation resembling gliosis or tumor-like growth. Combining gene-specific epigenetic modulation (e.g., Dnmt3a CRISPRi) with controlled activation of pro-neural transcription factors could promote more complete lineage conversion while preserving retinal architecture.

In addition, if induced epigenetic alterations persist over time, they may lead to epigenomic instability. Future studies could employ inducible CRISPR-based systems to restrict gene modulation to defined temporal windows, thereby ensuring that regenerative outcomes enhance repair without compromising retinal homeostasis. Such refined control may ultimately provide safer and more effective therapeutic avenues for retinal diseases.

## Supporting information

**S1 Fig. Determination of transfection efficiency.** Data obtained from the quantification of Cas9 positive cells after 48 hours post transfection with a previous 24 h exposition to a high dose of puromycin.
(XLSX)

**S2 Fig. Plasmid construct validation.** A) Representative immunofluorescence imagen of CTRL and DNMT3a KD groups to validate Müller cell transfection protocol, determined by Cas9 protein (green) for plasmid and glutamine synthetase (GS, red) for Müller cells delimitation. Nuclei were stained with DAPI (blue). Scale bar = 20 µm, 40X. B) Schematic representation of the general mechanism of action of CRISPRi systems. C) Representative agarose gel for plasmid digestions. D) Representative agarose gel for conventional PCR products for guide RNA (sgRNA) ligation into plasmid vector + : positive control to U6 promoter, sgRNA and gRNA scaffold fragment amplification; -: non template control; MWM: molecular weight marker. E) Schematic representation of PCRs amplifications presented in panel D, generated with SnapGene.
(TIFF)

**S1 Data. Determination of transfection efficiency.**
(XLSX)

## Acknowledgments

The authors would like to thank Dr. Rene Garduño Gutierrez for his assistance in confocal imaging experiments and guidance and all the lab members for helpful discussions.

## Author contributions

**Conceptualization:** Rebeca Victoria-Chavez, Monica Lamas.

**Data curation:** Rebeca Victoria-Chavez, Monica Lamas.

**Formal analysis:** Monica Lamas.

**Funding acquisition:** Monica Lamas.

**Investigation:** Rebeca Victoria-Chavez, Monica Lamas.

**Methodology:** Rebeca Victoria-Chavez, Monica Lamas.

**Project administration:** Monica Lamas.

**Resources:** Monica Lamas.

**Software:** Monica Lamas.

**Supervision:** Monica Lamas.

**Validation:** Monica Lamas.

**Visualization:** Monica Lamas.

**Writing – original draft:** Rebeca Victoria-Chavez, Monica Lamas.

**Writing – review & editing:** Rebeca Victoria-Chavez, Monica Lamas.

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
