## [Decision Letter · Decision Letter 0]

6 Sep 2025

Dear Dr. Lamas,

Thank you for submitting your manuscript to PLOS ONE. After careful consideration, we feel that it has merit but does not fully meet PLOS ONE’s publication criteria as it currently stands. Therefore, we invite you to submit a revised version of the manuscript that addresses the points raised during the review process.

We look forward to receiving your revised manuscript.

Kind regards,

Peng Zhang, Ph.D.

Academic Editor

PLOS ONE

Journal Requirements:

This work was supported by Conahcyt grant A1-S-25777 and Velux Stiftung project number 1852. A PhD fellowship from Conahcyt to R. V-C is also acknowledged.

Reviewers' comments:

Reviewer's Responses to Questions

**Comments to the Author**

1. Is the manuscript technically sound, and do the data support the conclusions?

Reviewer #1: Yes

Reviewer #2: Yes

Reviewer #3: Yes

2. Has the statistical analysis been performed appropriately and rigorously?

Reviewer #1: Yes

Reviewer #2: Yes

Reviewer #3: Yes

3. Have the authors made all data underlying the findings in their manuscript fully available?

Reviewer #1: Yes

Reviewer #2: Yes

Reviewer #3: Yes

4. Is the manuscript presented in an intelligible fashion and written in standard English?

Reviewer #1: Yes

Reviewer #2: Yes

Reviewer #3: Yes

Reviewer #1: This is an exciting and important paper that identifies DNMT3A inhibition as a novel epigenetic lever to promote Müller glia (MG) reprogramming in mouse retina, complementing prior work from the same group showing that TET3 induction drives similar neuron-like phenotypes. The data convincingly demonstrate increased neuronal marker expression (βIII-tubulin), progenitor signatures (LIN28, Nestin), and enhanced proliferation and migration, pointing toward a push–pull epigenetic model (DNMT3A as a brake, TET3 as an accelerator) that could help unlock mammalian retinal regeneration akin to the robust MG-mediated repair seen in zebrafish.

Major Comments:

To fully contextualize the significance of these findings, the authors should add a comparative table contrasting zebrafish MG regeneration, prior mammalian MG reprogramming strategies (e.g., Ascl1 + HDACi, TF cocktails), and the current DNMT3A-targeting approach, highlighting differences in injury triggers, molecular drivers, epigenetic states, and regenerative outcomes.

For future studies, it would be valuable to discuss the potential of small-molecule DNMT inhibitors, siRNA approaches, or TET inhibitors (e.g., Bobcat339) as orthogonal validation strategies, as well as the importance of direct epigenomic profiling (DNA methylation/5hmC, ATAC-seq) and testing within injury and immune contexts where microglia are known to modulate MG reprogramming.

With these additions, this work will not only stand as a strong mechanistic advance but also serve as a roadmap for translational strategies aimed at inducing regeneration in the mammalian retina.

Minor comment: page 22 (Kiy should be Kit).

Reviewer #2: This is a well-executed and potentially impactful study that addresses a critical barrier in mammalian retinal regeneration. The use of a CRISPRi system for transient and targeted epigenetic modulation is sophisticated and appropriate. The idea that DNMT3a acts as a repressive barrier counteracting initial pro-regenerative demethylation events is innovative and well-supported by the data. The findings are compelling and suggest DNMT3a is a significant epigenetic brake on Müller glia (MG) plasticity.

Major Points for Revision:

1.The title and abstract emphasize "transient inhibition," but the method is a knockdown via transfection. While the effect may not be permanent, "transient" could be misinterpreted. Clarify the duration of the knockdown effect (shown to be ~3-5 days) and justify the terminology or consider rephrasing (e.g., "targeted knockdown").

2.A discussion on the potential and challenges of translating this in vitro CRISPRi approach to an in vivo mammalian model is crucial. This should be added to the Discussion.

3.Several figures (as placeholder text) are referenced but not available for review. The figure legends must be exceptionally clear. Ensure all panels are labeled (A, B, C, etc.), statistical test details (n numbers, exact p-values where possible) are included, and scale bars are visible on the final images.

Specific Comments by Section

1. Title: It is accurate but slightly long.

2. Introduction Consider adding a sentence explicitly stating the knowledge gap: e.g., "While the role of demethylation has been explored, the specific contribution of de novo methyltransferases like DNMT3a in restricting mammalian MG plasticity remains unclear."

3. Discussion The co-expression of glial and neuronal markers (Fig 7) needs to be discussed in more depth. What does this mean for the functionality of these cells? Could this be an intermediate state that requires additional cues for full maturation? This is a key point for the field.

As noted above, a paragraph discussing the practical challenges and future possibilities of applying this strategy in vivo (e.g., using AAV-delivered CRISPRi) is necessary. Would transient inhibition be sufficient to stimulate regeneration in a model of retinal degeneration?

4. Materials and Methods The protocol for obtaining pure MG cultures is standard. However, stating the final percentage of GS-positive cells (e.g., >95%) from an immunofluorescence validation would strengthen the methodology. Please include the efficiency of electroporation/transfection, as determined by Cas9 immunofluorescence. This is important for interpreting the % of cells showing effects.

Reviewer #3: 1. As mentioned in the introduction, DNMT3a, has been identified as a crucial regulator in optic nerve regeneration, and its downregulation has been shown to induce passive demethylation at neurogenic gene loci in different cell types. In addition, a recent review (Zhou et al., 2025; Advancements in Müller Glia Reprogramming: Pioneering Approaches for Retinal Neuron Regeneration) emphasizes that DNA demethylation is crucial for initiating Müller glia reprogramming. These observations raise questions regarding the distinction of this study from previous works. The authors should clarify more explicitly what is novel about their study.

2. Off-target effects of CRISPRi are only minimally addressed. The authors should point out that the knockdown of DNMT3b and DNMT1 has not been evaluated beyond qPCR.

3. The results indicate that DNMT3a knockdown induced βIII-tubulin expression in the cells, while glial markers were also retained. This suggests only partial reprogramming of Müller glia cells. The authors should address this as a limitation and discuss possible solutions as future directions.

4. The use of the CRISPRi system for the transient downregulation of DNMT3a is elegant and specific, but technically demanding. The authors should explain clearly the advantages and disadvantages of this system over pharmacological inhibition by small-molecule DNMT inhibitors, which are easier to apply.

5. Although no significant negative impact of transient DNMT3a inhibition by the CRISPRi system on cell viability was observed in vitro, the possible safety concerns should be briefly discussed. These include the possibility of aberrant or incomplete reprogramming, deregulation of additional pathways, uncontrolled proliferation or oncogenic transformation, and instability of epigenetic state. Given the therapeutic aim of this study, the authors should acknowledge the associated risks and highlight that further in vivo safety and functional studies are essential before considering any clinical application. A brief discussion of these points would strengthen the therapeutic relevance of the manuscript.

6. The neural differentiation analysis focuses on morphology and marker expression (βIII-tubulin), without functional validation. At minimum, the authors should acknowledge the limitation of relying solely on this early neuronal marker and emphasize that additional differentiation markers or functional assays will be required to confirm neuronal identity and functionality.

**Do you want your identity to be public for this peer review?** For information about this choice, including consent withdrawal, please see our Privacy Policy

Reviewer #1: **Yes: ** Douglas M Ruden

Reviewer #2: No

Reviewer #3: No

---

## [Author Response · Author response to Decision Letter 1]

17 Oct 2025

Following you will find the poin-by-point response to the academic editor.

Journal Requirements:

Thank you.

This work was supported by Conahcyt grant A1-S-25777 and Velux Stiftung project number 1852. A PhD fellowship from Conahcyt to R. V-C is also acknowledged.

The sentence has been included.

Revised.

4. If the reviewer comments include a recommendation to cite specific previously published works, please review and evaluate these publications to determine whether they are relevant and should be cited.

No citations were suggested by the reviewers. However, nine new relevant cites have been included to complete the Introduction and Discussion Sections.

Response to the Reviewers

Reviewer #1:

Major Comments:

To fully contextualize the significance of these findings, the authors should add a comparative table contrasting zebrafish MG regeneration, prior mammalian MG reprogramming strategies (e.g., Ascl1 + HDACi, TF cocktails), and the current DNMT3A-targeting approach, highlighting differences in injury triggers, molecular drivers, epigenetic states, and regenerative outcomes.

For future studies, it would be valuable to discuss the potential of small-molecule DNMT inhibitors, siRNA approaches, or TET inhibitors (e.g., Bobcat339) as orthogonal validation strategies, as well as the importance of direct epigenomic profiling (DNA methylation/5hmC, ATAC-seq) and testing within injury and immune contexts where microglia are known to modulate MG reprogramming.

With these additions, this work will not only stand as a strong mechanistic advance but also serve as a roadmap for translational strategies aimed at inducing regeneration in the mammalian retina.

We sincerely thank you for your thoughtful and constructive review, which has significantly improved our manuscript. In response to your suggestion, we have now included a comparative table (Table 1).

To comply with your suggestions we have included the following paragraph in the Discussion section toghether with several new references:

“Dnmt3a CRISPR-dCas9-mediated downregulation may represent a promising strategy to prolong the transient progenitor state of MG and enhance their neurogenic potential and retinal regenerative capacity in mammals. Importantly, this approach could complement several reprogramming strategies previously applied in both regenerative and non-regenerative species (Table 1). Orthogonal validation methods, such as small-molecule epigenetic modulators (73), siRNAs (27), or CRISPR-CasRx (72), would be invaluable to reinforce the robustness and significance of these findings.

For these observations to have a physiological impact, it is essential to translate these studies into an in vivo system, where the interaction of Müller cells with all retinal and immune cells, including microglia, can be properly taken into account. Although complex, this is a feasible task. The CRISPR/Cas system has been successfully adapted to target retinal cells in vivo, either through electroporation (74, 75) or by adeno-associated virus 2 (AAV2)-mediated deliver, an approach that holds greater promise for future clinical applications (76).”

Minor comment: page 22 (Kiy should be Kit).

Corrected.

Thank you very much for all your suggestions.

Reviewer #2:

Major Points for Revision:

1.The title and abstract emphasize "transient inhibition," but the method is a knockdown via transfection. While the effect may not be permanent, "transient" could be misinterpreted. Clarify the duration of the knockdown effect (shown to be ~3-5 days) and justify the terminology or consider rephrasing (e.g., "targeted knockdown").

We have incorporated your suggestion by revising our title to “Targeted knockdown of DNA methyltransferase 3a (DNMT3a) unlocks Dedifferentiation and Neurogenic Potential in Mouse Retinal Müller Glia”; and by clarifying the duration of the knockdown effect in the text.

2.A discussion on the potential and challenges of translating this in vitro CRISPRi approach to an in vivo mammalian model is crucial. This should be added to the Discussion.

Taking into consideration the reviewers´ suggestions we have now enriched our Discussion with the following paragraph:

“For these observations to have a physiological impact, it is essential to translate these studies into an in vivo system, where the interaction of Müller cells with all retinal and immune cells, including microglia, can be properly taken into account. Although complex, this is a feasible task. The CRISPR/Cas system has been successfully adapted to target retinal cells in vivo, either through electroporation (74, 75) or by adeno-associated virus 2 (AAV2)-mediated deliver, an approach that holds greater promise for future clinical applications (76). Although they offer high efficiency and specificity, when compared for example with the available pharmacological DNMT inhibitors, CRISPR/Cas9 gene editing technologies are not without limitations. These include potential off-target effects, variability in editing efficiency across cell types, immune responses against Cas proteins or viral vectors, and challenges in achieving precise spatial and temporal control of gene modulation within the retina to achieve sufficient regeneration. Resolving these issues may open new therapeutic avenues for retinal diseases.”

3.Several figures (as placeholder text) are referenced but not available for review. The figure legends must be exceptionally clear. Ensure all panels are labeled (A, B, C, etc.), statistical test details (n numbers, exact p-values where possible) are included, and scale bars are visible on the final images.

We have revised our document to avoid errors. Thank you.

Specific Comments by Section

1. Title: It is accurate but slightly long.

We have revised our title to “Targeted knockdown of DNA methyltransferase 3a (DNMT3a) Unlocks dedifferentiation and Neurogenic Potential in Mouse Retinal Müller Glia”

2. Introduction Consider adding a sentence explicitly stating the knowledge gap: e.g., "While the role of demethylation has been explored, the specific contribution of de novo methyltransferases like DNMT3a in restricting mammalian MG plasticity remains unclear."

Thank you very much. We have added this sentence in page 4.

3. Discussion The co-expression of glial and neuronal markers (Fig 7) needs to be discussed in more depth. What does this mean for the functionality of these cells? Could this be an intermediate state that requires additional cues for full maturation? This is a key point for the field.

Indeed, this is a key point for the field. However, there is not a lot of information as most of the articles on the area do not address the co-expression of glial and neuronal markers. We wanted to include this figure to start adding to the evidence, convinced that more examples are going to be reported. However, it is still too early for us to fully discuss possible physiological implications of these observations. We have now included in the text our “future direction”. It is an ongoing but promising field.

This is the sentence that we have now included in the Discussion section:

“Further electrophysiological characterization of βIII-tubulin–expressing cells will be required to confirm their neuronal functionality. Such analyses could also help determine whether, for instance, temporal control of DNMT3a expression might extend the window of neuronal differentiation, thereby leading not only to an increased number of neurons but also to the acquisition of a fully functional neuronal phenotype.”

As noted above, a paragraph discussing the practical challenges and future possibilities of applying this strategy in vivo (e.g., using AAV-delivered CRISPRi) is necessary. Would transient inhibition be sufficient to stimulate regeneration in a model of retinal degeneration?

Here is the paragraph that we included:

“For these observations to have a physiological impact, it is essential to translate these studies into an in vivo system, where the interaction of Müller cells with all retinal and immune cells, including microglia, can be properly taken into account. Although complex, this is a feasible task. The CRISPR/Cas system has been successfully adapted to target retinal cells in vivo, either through electroporation (74, 75) or by adeno-associated virus 2 (AAV2)-mediated deliver, an approach that holds greater promise for future clinical applications (76). Although they offer high efficiency and specificity, when compared for example with the available pharmacological DNMT inhibitors, CRISPR/Cas9 gene editing technologies are not without limitations. These include potential off-target effects, variability in editing efficiency across cell types, immune responses against Cas proteins or viral vectors, and challenges in achieving precise spatial and temporal control of gene modulation within the retina to achieve sufficient regeneration. Resolving these issues may open new therapeutic avenues for retinal diseases.”

4. Materials and Methods The protocol for obtaining pure MG cultures is standard. However, stating the final percentage of GS-positive cells (e.g., >95%) from an immunofluorescence validation would strengthen the methodology. Please include the efficiency of electroporation/transfection, as determined by Cas9 immunofluorescence. This is important for interpreting the % of cells showing effects.

This information has been included in the text:

“More than 95% of the cells were immunopositive to GS (not shown).”

“After puromycin selection more than 90% of the cells were immunopositive to Cas9.”

Thank you very much for your help.

Reviewer #3:

1. As mentioned in the introduction, DNMT3a, has been identified as a crucial regulator in optic nerve regeneration, and its downregulation has been shown to induce passive demethylation at neurogenic gene loci in different cell types. In addition, a recent review (Zhou et al., 2025; Advancements in Müller Glia Reprogramming: Pioneering Approaches for Retinal Neuron Regeneration) emphasizes that DNA demethylation is crucial for initiating Müller glia reprogramming. These observations raise questions regarding the distinction of this study from previous works. The authors should clarify more explicitly what is novel about their study.

Indeed, DNMT3 has been identified as a crucial regulator of retinal ganglion cell regeneration and axonal growth. However, no role for this enzyme has yet been described in Müller glia. To explicitly highlight the novelty of our study, we have included the following sentence in the Introduction:

“While the role of demethylation has been explored, the specific contribution of de novo methyltransferases like DNMT3a in restricting mammalian MG plasticity remains unclear.”

2. Off-target effects of CRISPRi are only minimally addressed. The authors should point out that the knockdown of DNMT3b and DNMT1 has not been evaluated beyond qPCR.

We have now precisely specified this in page 6.

“Specificity of the effect of the inhibition system on Dnmt3a expression was confirmed by RT-qPCR evaluation of the expression of Dnmt3b and Dnmt1 under the same experimental conditions. No significant effects were observed (Fig. 1 B,C), although possible effects of Dnmt3b and Dnmt1 were not evaluated beyond qPCR.”

3. The results indicate that DNMT3a knockdown induced βIII-tubulin expression in the cells, while glial markers were also retained. This suggests only partial reprogramming of Müller glia cells. The authors should address this as a limitation and discuss possible solutions as future directions.

Indeed, this is a key point for the field. However, there is not a lot of information as most of the articles on the area do not address the co-expression of glial and neuronal markers. We wanted to include this figure to start adding to the evidence, convinced that more examples are going to be reported. However, it is still too early for us to fully discuss possible physiological implications of these observations. We have now included in the text our “future direction”. It is an ongoing but promising field.

This is the sentence that we have now included in the Discussion section:

“Further electrophysiological characterization of βIII-tubulin–expressing cells will be required to confirm their neuronal functionality. Such analyses could also help determine whether, for instance, temporal control of DNMT3a expression might extend the window of neuronal differentiation, thereby leading not only to an increased number of neurons but also to the acquisition of a fully functional neuronal phenotype.”

4. The use of the CRISPRi system for the transient downregulation of DNMT3a is elegant and specific, but technically demanding. The authors should explain clearly the advantages and disadvantages of this system over pharmacological inhibition by small-molecule DNMT inhibitors, which are easier to apply.

5. Although no significant negative impact of transient DNMT3a inhibition by the CRISPRi system on cell viability was observed in vitro, the possible safety concerns should be briefly discussed. These include the possibility of aberrant or incomplete reprogramming, deregulation of additional pathways, uncontrolled proliferation or oncogenic transformation, and instability of epigenetic state. Given the therapeutic aim of this study, the authors should acknowledge the associated risks and highlight that further in vivo safety and functional studies are essential before considering any clinical application. A brief discussion of these points would strengthen the therapeutic relevance of the manuscript.

Regarding these two comments we have included this new information in the Dicussion section:

“For these observations to have a physiological impact, it is essential to translate these studies into an in vivo system, where the interaction of Müller cells with all retinal and immune cells, including microglia, can be properly taken into account. Although complex, this is a feasible task. The CRISPR/Cas system has been successfully adapted to target retinal cells in vivo, either through electroporation (74, 75) or by adeno-associated virus 2 (AAV2)-mediated deliver, an approach that holds greater promise for future clinical applications (76). Although they offer high efficiency and specificity, when compared for example with the available pharmacological DNMT inhibitors, CRISPR/Cas9 gene editing technologies are not without limitations. These include potential off-target effects, variability in editing efficiency across cell types, immune responses against Cas proteins or viral vectors, and challenges in achieving precise spatial and temporal control of gene modulation within the retina to achieve sufficient regeneration. Resolving these issues may open new therapeutic avenues for retinal diseases.”

6. The neural differentiation analysis focuses on morphology and marker expression (βIII-tubulin), without functional validation. At minimum, the authors should acknowledge the limitation of relying solely on this early neuronal marker and emphasize that additional differentiation markers or functional assays will be required to confirm neuronal identity and functionality.

We agree. We have included a statement indicating “Further electrophysiological characterization of βIII-tubulin-expressing cells will be required to confirm neuronal functionality”.

Thank you very much for your review.

---

## [Decision Letter · Decision Letter 1]

27 Oct 2025

Dear Dr. Lamas,

Thank you for submitting your manuscript to PLOS ONE. After careful consideration, we feel that it has merit but does not fully meet PLOS ONE’s publication criteria as it currently stands. Therefore, we invite you to submit a revised version of the manuscript that addresses the points raised during the review process.

We look forward to receiving your revised manuscript.

Kind regards,

Peng Zhang, Ph.D.

Academic Editor

PLOS ONE

Journal Requirements:

Reviewers' comments:

Reviewer's Responses to Questions

**Comments to the Author**

Reviewer #1: All comments have been addressed

Reviewer #2: All comments have been addressed

Reviewer #3: (No Response)

2. Is the manuscript technically sound, and do the data support the conclusions?

Reviewer #1: Yes

Reviewer #2: Yes

Reviewer #3: Yes

3. Has the statistical analysis been performed appropriately and rigorously?

Reviewer #1: Yes

Reviewer #2: Yes

Reviewer #3: Yes

4. Have the authors made all data underlying the findings in their manuscript fully available?

Reviewer #1: Yes

Reviewer #2: Yes

Reviewer #3: Yes

5. Is the manuscript presented in an intelligible fashion and written in standard English?

Reviewer #1: Yes

Reviewer #2: Yes

Reviewer #3: Yes

Reviewer #1: The paper is much improved and the inclusion of a table comparing zebrafish and mouse 3a inhibition studies is helpful. I only have a couple of minor corrections in the table: dcas9 should be dCas9 - and explain what CasRx and dCas9 are in the table or text.

Reviewer #2: (No Response)

Reviewer #3: The authors have comprehensively responded to the reviewers’ comments, and the manuscript has been substantially improved. However, several aspects would still benefit from further clarification.

1. The novelty of the study relative to previous works on DNMT3A-related reprogramming should be explained more explicitly. What distinguishes the present CRISPRi-based approach from prior demethylation strategies? Is the CRISPRi system preferable to previously published methods, and if so, why?

2. The authors should discuss the potential risks associated with partial reprogramming of the Müller glia cells, along with the possible solutions in the future.

3. As mentioned in the previous review round, the assessment of neuronal differentiation relies mainly on the morphology and the expression of βIII-tubulin (an early neuronal marker). To confirm neuronal identity and functionality, the authors should examine the expression of additional markers associated with the later stages of differentiation, like MAP2 and NEFL. Electrophysiological characterization would also be helpful to verify the final maturation of the neurons.

**Do you want your identity to be public for this peer review?** For information about this choice, including consent withdrawal, please see our Privacy Policy

Reviewer #1: **Yes: ** Douglas Ruden

Reviewer #2: No

Reviewer #3: No

---

## [Author Response · Author response to Decision Letter 2]

11 Nov 2025

Reviewers' Response

Reviewer #1: The paper is much improved and the inclusion of a table comparing zebrafish and mouse 3a inhibition studies is helpful. I only have a couple of minor corrections in the table: dcas9 should be dCas9 - and explain what CasRx and dCas9 are in the table or text.

The typo has been corrected and information about CasRx (Cas13d) has been included in the text, thank you.

In line 325 of the Results section, dCas9 is described in the text as a “catalitically inactive Cas9”.

“These sgRNAs were incorporated into CRISPR interference systems (CRISPRi) utilizing a catalytically inactive Cas9 (dCas9) fused to the Krüppel-associated box (KRAB) repressor domain (43) (S1). This configuration enables targeted transient transcriptional repression of Dnmt3a expression in transfected cells.”

In line 518 of the Discussion, brief information about CasRx is provided:

“Orthogonal validation methods, such as small-molecule epigenetic modulators (73), siRNAs (27), or the recently characterized and optimized CRISPR protein family Cas13, also known as CRISPR-CasRx (72), would be invaluable to reinforce the robustness and significance of these findings”

Reviewer #3: The authors have comprehensively responded to the reviewers’ comments, and the manuscript has been substantially improved. However, several aspects would still benefit from further clarification.

1. The novelty of the study relative to previous works on DNMT3A-related reprogramming should be explained more explicitly. What distinguishes the present CRISPRi-based approach from prior demethylation strategies? Is the CRISPRi system preferable to previously published methods, and if so, why?

2. The authors should discuss the potential risks associated with partial reprogramming of the Müller glia cells, along with the possible solutions in the future.

Combining these two questions, we have added this information in the Dsicussion Section:

“CRISPRi systems offer high efficiency and specificity, when compared for example with the available non-specific pharmacological DNMT inhibitors (e.g., 5-azacytidine or RG108) which broadly reduce methylation across the genome and may indirectly affect multiple cellular pathways, often leading to cytotoxicity or limited temporal control. The use of a catalytically inactive Cas9 (dCas9) fused to the KRAB repressor domain enabled precise, locus-specific and transient transcriptional repression of Dnmt3a, simultaneously minimizing off-target disturbances and preserving the necessary temporal reversibility of the epigenetic alteration. In addition, these procedures are scalable and should permit the evaluation of the coordinated regulation of other transcription factors.

However, CRISPR/Cas9 gene editing technologies are not without limitations, and resolving such limitations could overcome some of the potential risks that partial reprogramming of Müller glia carry. For example, incomplete or unstable reprogramming may lead to cellular heterogeneity, with subpopulations that co-express glial and neuronal markers but fail to achieve full neuronal functionality. Such intermediate states could result in aberrant signaling, impaired glial support functions, or even uncontrolled proliferation resembling gliosis or tumor-like growth. Combining gene-specific epigenetic modulation (e.g., Dnmt3a CRISPRi) with controlled activation of pro-neural transcription factors could promote more complete lineage conversion while preserving retinal architecture.

In addition, if induced epigenetic alterations persist over time, they may lead to epigenomic instability. Future studies could employ inducible CRISPR-based systems to restrict gene modulation to defined temporal windows, thereby ensuring that regenerative outcomes enhance repair without compromising retinal homeostasis. Such refined control may ultimately provide safer and more effective therapeutic avenues for retinal diseases.”

3. As mentioned in the previous review round, the assessment of neuronal differentiation relies mainly on the morphology and the expression of βIII-tubulin (an early neuronal marker). To confirm neuronal identity and functionality, the authors should examine the expression of additional markers associated with the later stages of differentiation, like MAP2 and NEFL. Electrophysiological characterization would also be helpful to verify the final maturation of the neurons.

Yes, we fully agree with you, and we are currently developing that project. We believe that immunological characterization alone is insufficient and would leave important questions unresolved. Therefore, we are conducting an extensive electrophysiological analysis. Müller cells pose certain challenges for patch-clamp recordings, whereas progenitor cells and βIII-tubulin–expressing cells are more readily analyzed and display distinct electrophysiological patterns. Our preliminary observations reveal a considerable degree of heterogeneity in the electrophysiological responses of these cells. As you rightly note, this is a critical issue, and we prefer not to draw premature conclusions at this stage. We have revised the Discussion to clarify this point and hope to be granted the time necessary to fully address this important aspect.

“Further genotypic characterization of βIII-tubulin–expressing cells, through the evaluation of their immunoreactivity to MAP2 or NeuroD1 antibodies, as well as electrophysiological assessment of their intrinsic properties, is required to confirm their neuronal functionality”.

Thank you very much for your review.

---

## [Decision Letter · Decision Letter 2]

16 Nov 2025

Targeted knockdown of DNA methyltransferase 3a (DNMT3a) unlocks dedifferentiation and neurogenic potential in mouse retinal Müller glia

PONE-D-25-43090R2

Dear Dr. Lamas,

We’re pleased to inform you that your manuscript has been judged scientifically suitable for publication and will be formally accepted for publication once it meets all outstanding technical requirements.

Kind regards,

Peng Zhang, Ph.D.

Academic Editor

PLOS ONE

Additional Editor Comments (optional):

Reviewers' comments:

Reviewer's Responses to Questions

**Comments to the Author**

Reviewer #1: All comments have been addressed

Reviewer #3: All comments have been addressed

2. Is the manuscript technically sound, and do the data support the conclusions?

Reviewer #1: Yes

Reviewer #3: Yes

3. Has the statistical analysis been performed appropriately and rigorously?

Reviewer #1: Yes

Reviewer #3: N/A

4. Have the authors made all data underlying the findings in their manuscript fully available?

Reviewer #1: Yes

Reviewer #3: Yes

5. Is the manuscript presented in an intelligible fashion and written in standard English?

Reviewer #1: Yes

Reviewer #3: Yes

Reviewer #1: All of my concerns have been adequately addressed in this interesting and timely paper. The authors included a table that I requested and the paper is much improved.

Reviewer #3: (No Response)

**Do you want your identity to be public for this peer review?** For information about this choice, including consent withdrawal, please see our Privacy Policy

Reviewer #1: **Yes: ** Douglas Ruden

Reviewer #3: No

---

## [Editor Report · Acceptance letter]

PONE-D-25-43090R2

PLOS One

Dear Dr. Lamas,

I'm pleased to inform you that your manuscript has been deemed suitable for publication in PLOS One. Congratulations! Your manuscript is now being handed over to our production team.

Kind regards,

on behalf of

Prof. Peng Zhang

Academic Editor

PLOS One